# Sim2Real-Fire: A Multi-modal Simulation Dataset for Forecast and Backtracking of Real-world Forest Fire

**Yanzhi Li**[1,2]    **Keqiu Li**[1]    **Guohui Li**[2]    **Zumin Wang**[3]    **Changqing Ji**[3]

**Lubo Wang**[1]    **Die Zuo**[1]    **Qing Guo**[4]    **Feng Zhang**[5]    **Manyu Wang**[1*]    **Di Lin**[1*]

[1]Tianjin University, China
[2]Tianjin Fire Science and Technology Research Institute of MEM, China
[3]Dalian University, China
[4]CFAR and IHPC, Agency for Science, Technology and Research (A*STAR), Singapore
[5]Hebei University, China
wmy22@tju.edu.cn  di.lin@tju.edu.cn

## Abstract

The latest research on wildfire forecast and backtracking has adopted AI models, which require a large amount of data from wildfire scenarios to capture fire spread patterns. This paper explores using cost-effective simulated wildfire scenarios to train AI models and apply them to the analysis of real-world wildfire. This solution requires AI models to minimize the Sim2Real gap, a brand-new topic in the fire spread analysis research community. To investigate the possibility of minimizing the Sim2Real gap, we collect the Sim2Real-Fire dataset that contains 1M simulated scenarios with multi-modal environmental information for training AI models. We prepare 1K real-world wildfire scenarios for testing the AI models. We also propose a deep transformer, S2R-FireTr, which excels in considering the multi-modal environmental information for forecasting and backtracking the wildfire. S2R-FireTr surpasses state-of-the-art methods in real-world wildfire scenarios.

## 1 Introduction

The frequency and intensity of extreme weather events, such as high temperatures and droughts, have escalated. This escalation has increased the frequency and scale of forest fires, rendering fire extinguishing a formidable challenge. An accurate and real-time forest fire spread forecast is imperative for organizing evacuations and commanding rescue operations. On the other hand, forest fire backtracking helps identify high-risk ignition areas, assisting people in preventing potential fires.

Extensive studies have been conducted on forest fire spread forecast and backtracking. These studies have given rise to three categories of methods based on the empirical, physical, or artificial intelligence (AI) models. The empirical models [1; 2; 3; 4; 5; 6; 7] only capture the statistical correlation between observed fire data in real-world or the energy conservation, without considering the impact of physical rules like the conductive, convective, and radiative modes of heat transfer on the fire spread. The physical models [8; 9; 10; 11; 12] rely on the approximate physical rules, including the fluid dynamics, combustion, and heat transfer, to forecast the forest fire spread. Furthermore, they also consider the impact of environmental information about vegetation, atmosphere, and topography on fire spread. The research on the empirical and physical models leads to the emergence of an array of simulators of forest fire spread like BehavePlus [13], FARSITE [1], FIRETEC [14], WRF-SFIRE [12], and

---

[*]Corresponding author.

38th Conference on Neural Information Processing Systems (NeurIPS 2024) Track on Datasets and Benchmarks.

WFDS [9]. Given the environmental details and the current fire area, a simulator can predict the spread area at any moment. This is also known as the simulation process. However, the existing simulators fall short of utilizing the historical multi-modal data (e.g., satellite-view images of forest and spreading fire areas) of fire progression to forecast the future areas of fire spreading. These simulators also lack backtracking capability, which traces earlier fire states.

AI models [15; 16; 17; 18; 19; 20; 21; 22; 23; 24; 25; 26; 27; 28; 29; 30] can better leverage the history of fire data to forecast and backtrack the spread of forest fire. Mainly, AI models based on the deep neural networks [20; 24; 25; 28; 29; 30] perform excellently in forest fire forecast and backtracking. These models learn the spreading patterns of forest fire from a large amount of data about the temporal change of the environmental factors and forest fire areas. The environmental data and the wildfire images are usually captured by remote sensing satellites from real-world forests. They form the multi-modal data that requires significant labor costs for data collection and annotation for model training. Moreover, the satellites orbit the Earth rapidly, without continuously observing a particular forest fire area. During the period without observation by any satellite, the environmental and fire data are unavailable, contributing to the difficulty of using real-world data for model training. Though the Gazer satellite can provide temporally complete data, it makes data collection extraordinarily expensive. Therefore, collecting large-scale and temporally complete data at a low cost is critical.

The simulation process based on the empirical and physical models can be done quickly without needing substantial human effort for data collection and annotation. The natural idea is to use the data generated by the simulator, in chronological or reverse order, to train the forecasting or backtracking model primarily based on the data-hungry deep network. Given the multi-modal environmental data of vegetation, atmosphere, and topography, the simulator can generate data on wildfire areas that change over time. This effectively addresses the problem of missing data due to intermittent observation by satellite. Besides, people can let the forest fire start at any possible position in the simulated environment, producing diverse fire-spreading data for model training.

The above manner takes advantage of the empirical, physical, and AI models, using large-scale simulation data to train the AI-based forecast and backtracking models. However, this manner faces two problems relevant to the Sim2Real gap. First, as the simulator employs approximate physical rules to generate simulation data, it introduces simulation errors, resulting in the Sim2Real gap between simulated and real data. Second, different simulators predict fire changes based on empirical or physical models. Even with the same environmental conditions, these simulators may yield vastly different prediction results, further exacerbating the Sim2Real gap. Natural forests' exceptionally complex climate and terrain environments make it infeasible to verify whether the results obtained by different simulators are reliable. This further makes it challenging to eliminate erroneous fire simulation data. The incorrect simulation results introduce significant noise into model training.

We promote research on minimizing the Sim2Real gap and utilizing the simulation data to train AI models for the spread forecast and backtracking of wildfires in real-world forests. We collect the Sim2Real-Fire dataset that contains 1M virtual wildfire scenarios. These scenarios are produced by widely-used simulators, FARSITE [1], WFDS [9], and WRF-SFIRE [31]. We prepare the environmental data of vegetation, fuel, topography, weather, and fire areas for each scenario. These data are in the multi-modal format captured from the satellite's view. Sim2Real-Fire provides large-scale data for training the AI-based forecast and backtracking models. We collect 1K worldwide scenarios of wildfire in the natural forest. We select these real scenarios from publicly available satellite data. Compared to the existing datasets [19; 32; 33; 34; 35; 36], Sim2Real-Fire provides more large-scale and challenging data for testing the Sim2Real performances of AI models.

We also contribute a Sim2Real model, S2R-FireTr, based on a deep transformer network inspired by semantic segmentation methods [37; 38], for forest fire spread forecast and backtracking. We train S2R-FireTr on the simulation data. S2R-FireTr comprehensively captures the correlation between multi-modal data to alleviate the Sim2Real gap in network training. Moreover, S2R-FireTr can be trained on temporally incomplete data to enhance its forecast and backtracking capacities during testing on real-world data. We evaluate S2R-FireTr on the new Sim2Real-Fire dataset, surpassing the performances of state-of-the-art methods.

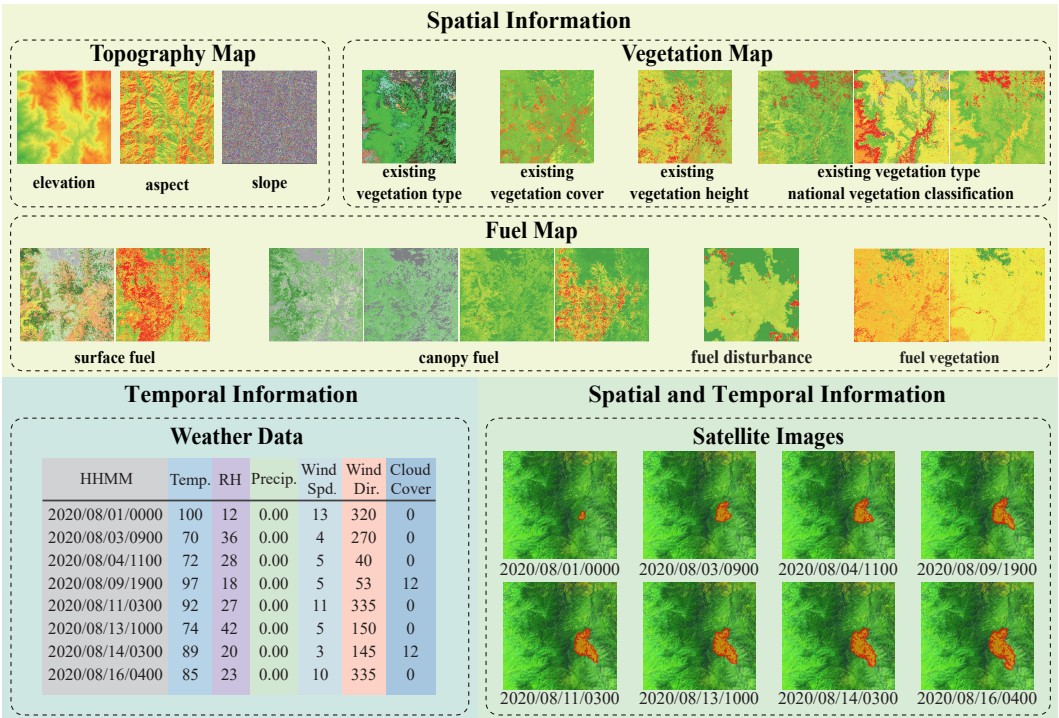

Figure 1: Topography, vegetation, fuel, weather, and the satellite data in the Sim2Real-Fire dataset.

## 2 Sim2Real-Fire Dataset

The Sim2Real-Fire dataset contains wildfire simulation and real-world data. The set includes 1M and 1K wildfire scenarios. Each scenario is associated with five data modalities of environmental information, including topography, vegetation, fuel maps, weather data, and satellite images with the annotated fire regions. We align these modalities spatially and temporally.

### 2.1 Data Modalities of Environmental Information

Figure 1 shows examples of topography, vegetation, fuel, weather, and satellite data.

**Topography Map** We follow the format of LANDFIRE [39] to make the topography map. Each topography map describes the landscape of a region of the United States, Canada, or Mexico from 2013 to 2023. It can be divided into three channels: landscape aspect, elevation, and slope. Aspect is the azimuth of sloping surfaces across a landscape. The elevation is the land elevation (meters) above mean sea level. The slope is the change in elevation over a specific area.

**Vegetation Map** We follow LANDFIRE to collect the vegetation map. Each vegetation map consists of the channels of existing vegetation type (EVT), existing vegetation cover (EVC), existing vegetation height (EVH), and existing vegetation type national vegetation classification(EVTNVC). EVT provides the classification of about 700 types of plants. EVC represents the vertical projection of a region's percent live canopy cover. EVH is the average height of the dominant vegetation. EVTNVC is an existing vegetation-type layer representing the distribution of vegetation groups based on the USNVC classification.

**Fuel Map** The fuel map has four channels: surface fuel (SF), canopy fuel (CF), fuel disturbance (FD), and fuel vegetation (FV). SF represents the fuel distribution of sizes and types. CF contains information about forest canopy cover, height, and density. FD integrates the individual disturbance of the burnable vegetation for modeling the fuel transition. FV is an adapted depiction of vegetation for converting continuous vegetation values into the fuel model.

**Weather Data** The weather data is tabular, collected from the Remote Automatic Weather Station [40]. Each table of weather data contains the temperature, relative humidity, precipitation, wind speed, wind direction, and cloud cover, which are recorded in the hourly sequence.

**Satellite Images**   We use the satellite image sequences of the fire regions, which are captured by Landsat-8 [41] and Sentinel-2 [42] satellites. Landsat-8 carries the Operational Land Imager and Thermal Infrared Sensor, orbiting the Earth every 99 minutes with a revisit period of 16 days. Sentinel-2 consists of 2A and 2B, with a revisit period of 10 and 5 days, respectively. Each image has the mask annotation of the fire regions.

## 2.2   Simulation Data

We use the simulation data to train and test the forecast and backtracking models. We prepare 1M virtual wildfire scenarios, each with five data modalities. The satellite image sequence of each scenario contains about 100 frames of fire spread. We divide these simulation data by 80%/20% to form the training and testing sets. Below, we introduce how to use the wildfire simulators (i.e., FARSITE [1], WFDS [9], and WRF-SFIRE [31]) to produce the virtual scenarios.

**Simulators**   FARSITE relies on the empirical model to simulate wildfires. The input to FARSITE consists of the topography, vegetation maps, and weather data. Given the above three modalities of environmental information, FARSITE simulates the fire spread represented by a sequence of mask annotations. We fuse the mask annotations of fire regions with the satellite images, thus approximating real-world fire regions with changing boundaries.

WFDS combines numerical methods and physical models to simulate wildfires. It allows tuning the speed and scope of fire spread by controllable parameters. WFDS can produce satellite images with fire regions and smoke, which show realism like real-world images.

Like WFDS, WRF-SFIRE takes input as the simulation process's topography, fuel maps, and weather data. It outputs detailed information on fire dynamics, including the rate of spread, fire intensity, and spatial extent hourly. Unlike FARSITE and WFDS, which assume the weather data are independent of the fire spread, WRF-SFIRE yields weather data that may be significantly affected by the fire spread, thus facilitating the analysis of fire-weather interactions.

**Simulated Masks of Fire Regions**   The simulator outputs a sequence of binary masks to represent the change of fire regions in each virtual scenario. To enrich the simulation data, we randomly jitter the initial location of the wildfire and other controllable parameters (e.g., the speed and scope of fire spread). Given an identical set of multi-modality environmental data (i.e., topography, vegetation, fuel maps, weather data, and satellite images), we employ the simulator with the jittered parameters to produce about 200 discrepant sequences of mask annotations.

## 2.3   Real-world Data

Apart from the masks of fire regions produced by the simulators, we collect 1K real-world wildfire scenarios from the satellite images. We recruit a group of human annotators to identify and label the fire regions. Each real-world scenario also has five modalities of environmental information. Each sequence has 2-10 satellite images. The real-world scenarios are used for model testing.

**Data Selection**   We select the satellite images collected by Landsat-8 and Sentinel-2, fuse the images of different wavebands, and eliminate the images without clearly observing the fire regions due to dense clouds or smoke occlusion. We keep the spatial resolution of these fused images to 30 meters to capture landscapes on the Earth. We convert the fused images into tones close to real images through pseudo-colorization.

**Data Annotation**   We import the fused satellite images into ArcGIS [43], allowing human annotators to label the binary masks of the wildfire regions. Each mask is a polygon. Each annotator uses NV5 GEOSPATIAL software [44] to identify fire areas automatically. We recruit 20 annotators to manage the labeling task. To guarantee the quality of the annotations, we require three annotators to cross-check every binary mask. People need to refine a mask disapproved by three annotators.

## 2.4   Dataset Statistics and Comparison

We list the basic information of different datasets for wildfire analysis in Table 1. The Sim2Real-Fire dataset contains 1M scenarios of wildfire spreading over the world. It provides five data modalities: topography, vegetation, fuel, weather, and satellite data. We divide these data modalities into two groups. The first group contains the topography, vegetation, fuel maps, and satellite images, which

| Name | Scenarios | Countries | Areas | Tasks | Period | Spatial Resolution | Temporal Resolution | Sim/Real | Modalities |
|---|---|---|---|---|---|---|---|---|---|
| GABAM [45] | 10K | Worldwide | 149,000,000 | Fire Behavior | 1990-2021 | 30m | 1year | Real Only | 1 |
| Fire Atlas [46] | 13M | Worldwide | 149,000,000 | Fire Behavior | 2003-2016 | 500m | 1day | Real Only | 1 |
| GlobFire [35] | 100M | Worldwide | 149,000,000 | Fire Behavior | 2001-2017 | 500m | 1day | Real Only | 1 |
| WildfireDB [47] | 17M | USA | 9,834,000 | Spread Forecast | 2012-2017 | 375m | 1day | Real Only | 4 |
| SeasFire Cube [48] | 20K | Worldwide | 149,000,000 | Burned Area Forecast | 2001-2021 | 27km | 8days | Real Only | 4 |
| Next Day Wildfire [33] | 18K | USA | 9,834,000 | Spread Forecast | 2012-2020 | 1km | 1day | Real Only | 4 |
| WildfireSpreadTS [36] | 607 | USA | 9,834,000 | Spread Forecast Spread Backtrack | 2018-2021 | 375m | 1day | Real Only | 4 |
| PT-FireSprd [32] | 80 | Portugal | 92,150 | Fire Behavior Spread Forecast Danger Forecast Burned Area Forecast | 2015-2021 | 1m-4km | 30mins-14hours30mins | Real Only | 1 |
| Mesogeos [49] | 25K | Mediterranean | 9,000,000 | Danger Forecast Burned Area Forecast | 2006-2022 | 1km | 1day | Real Only | 4 |
| MODIS Thermal Anomaly[50] | 40K | Worldwide | 149,000,000 | Danger Forecast Spread Forecast | 2000-2024 | 1km | 1day | Real Only | 3 |
| VIIRS Thermal Anomaly[51] | 40K | Worldwide | 149,000,000 | Danger Forecast Spread Forecast | 2012-2024 | 375m | 12hours | Real Only | 3 |
| NOAA HMS Fire[52] | 1K | North America | 24,710,000 | Danger Forecast | 2003-2024 | 2km | 1day | Real Only | 3 |
| NOAA HMS Smoke[53] | 1K | North America | 24,710,000 | Danger Forecast | 2005-2024 | 2km | 1day | Real Only | 1 |
| GOES Wildfire[54] | 1K | Western Hemisphere | 61,000,000 | Danger Forecast Spread Forecast Burned Area Forecast | 2017-2024 | 2km | 5mins | Real Only | 4 |
| NIFC Wildfire Perimeters[55] | 20K | USA | 9,834,000 | Spread Forecast Burned Area Forecast | 2000-2024 | 2km | 5mins | Real Only | 1 |
| **Sim2Real-Fire** | **1M** | **Worldwide** | **20,000,000** | **Spread Forecast Spread Backtrack** | **2013-2023** | **30m** | **1hour** | **Sim&Real** | **5** |

Table 1: Comparison with the related datasets for wildfire analysis.

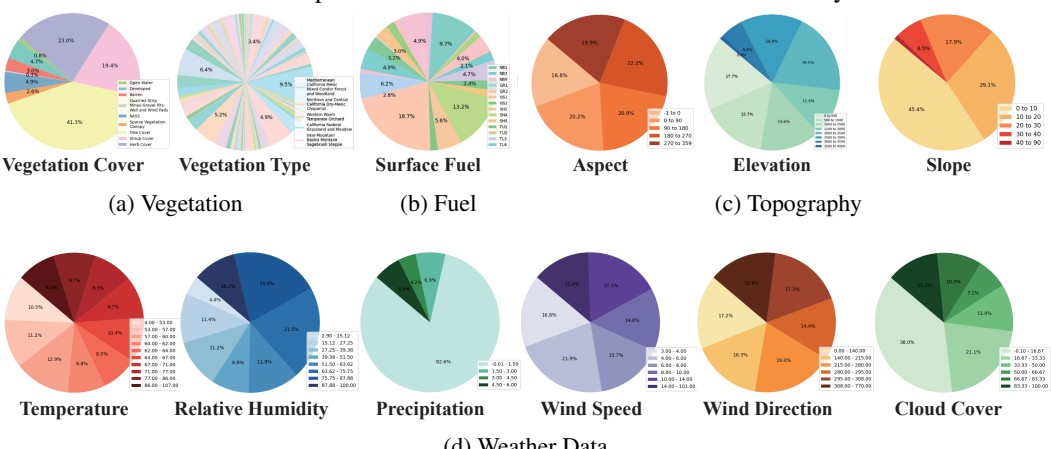

Figure 2: (a) Distribution of vegetation covers and types. (b) Distribution of fuel types. (c) Distribution topography data. (d) Distribution of weather data.

provide the spatial information of the wildfire scenarios. The second group contains sequential weather data and satellite images to capture the temporal dynamic of wildfire scenarios.

Compared to other datasets, Sim2Real-Fire offers richer environmental information across multiple modalities. The satellite images in this dataset have a spatial resolution of less than 30 meters for capturing wildfire scenarios, enabling more precise visualization of surrounding environments and fire areas. A key advantage of our dataset is its multi-modal hybrid data encompassing both simulated and real-life wildfire scenarios. The Sim2Real-Fire dataset is the first public dataset designed to support training forest fire forecast and backtracking models on simulation data and testing them on real-world data. The simulation data in this dataset was generated using various simulators (empirical, numerical, and physical models) to produce diverse fire scenarios, addressing the limitations that relying on a single simulator can introduce biases in model training.

Among the five modalities in the Sim2Real-Fire dataset, the vegetation and fuel maps provide the category-wise data. The topography map and weather data contain numerical data, which can be divided into several ranges. We report the proportions of the vegetation and fuel categories in Figure 2(a–b). The topography and weather data ranges are shown in Figure 2(c–d).

## 3 Architecture of S2R-FireTr

We regard the forecast or backtracking of forest fires as a temporal sequence-to-sequence translation task. Given the source sequence of with $T$ binary masks of forest fire areas as $\mathbf{S} \in \mathbb{R}^{H \times W \times T}$.

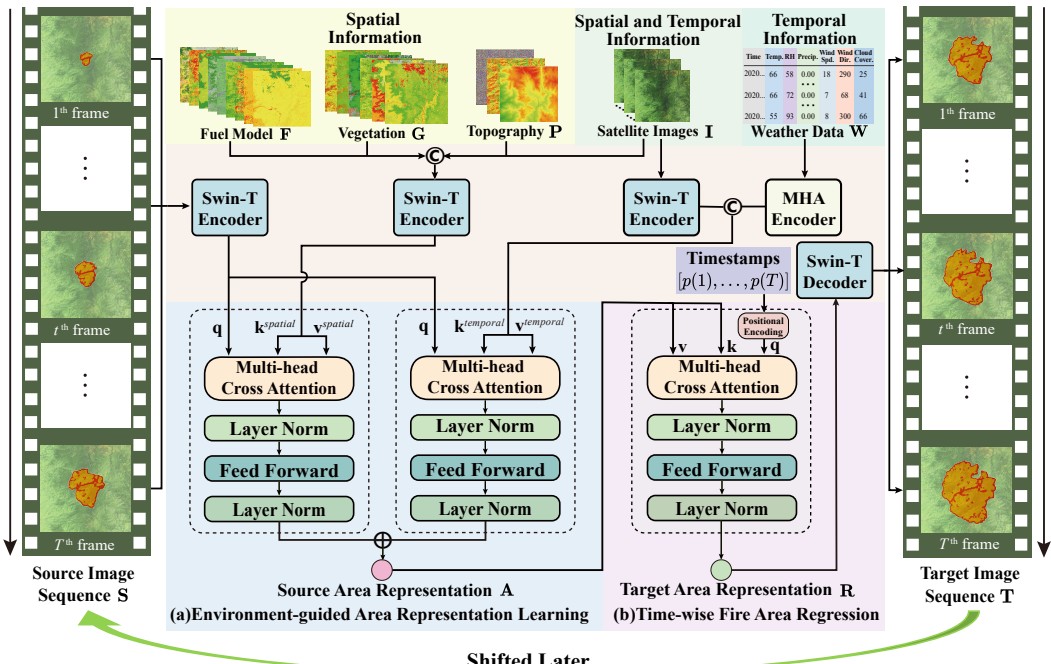

Figure 3: S2R-FireTr forecasts wildfires by predicting future target fire areas based on source fire areas. (a) During the environment-guided area representation learning, we input the source fire areas and multi-modal environmental information into the transformer to compute the source area presentation **A**. (b) During the time-wise fire area regression, we input the source area presentation **A** and the target timestamp into the transformer to compute the target area presentation **R** for predicting the target fire areas. **"Shifted Later"** means that we concatenate the source and target areas to predict later areas. Source and target areas can be interchanged, creating a pipeline for wildfire backtracking.

$H \times W$ indicates the spatial resolution of each image. The translation outputs the target sequence with $T$ binary masks $\mathbf{T} \in \mathbb{R}^{H \times W \times T}$ of forest fire areas. $\mathbf{T}$ represents the historical or future changes of the fire areas, temporally, in the forecast or backtracking task. We propose S2R-FireTr to accomplish the above translation. As illustrated in Figure 3, S2R-FireTr consists of the modules of **Environment-guided Area Representation Learning** and **Time-wise Fire Area Regression**.

**Environment-guided Area Representation Learning**  The first module of S2R-FireTr (Figure 3(a)) takes input as the source sequence of binary masks $\mathbf{S} \in \mathbb{R}^{H \times W \times T}$, which represents the known areas of forest fire within $T$ timestamps. We employ the satellite images $\mathbf{I} \in \mathbb{R}^{H \times W \times 3 \times T}$, the topography, vegetation, fuel maps $\mathbf{P}, \mathbf{G}, \mathbf{F} \in \mathbb{R}^{H \times W \times C}$, and the weather $\mathbf{W} \in \mathbb{R}^{C \times T}$ to learn the source area representation $\mathbf{A} \in \mathbb{R}^{H \times W \times C \times T}$ for the source sequence $\mathbf{S}$. $C$ indicates the channels.

We employ the dual cross-attention to learn the source area representation $\mathbf{A}$. In Eq. (1), the dual cross-attention considers the correlation between multi-modal data from the spatial and temporal perspectives. We compute the query vector $\mathbf{q} \in \mathbb{R}^{H \times W \times C \times T}$ based on the source sequence $\mathbf{S}$. We compute two sets of key and value vectors, $\mathbf{k}^{spatial}, \mathbf{v}^{spatial}, \mathbf{k}^{temporal}, \mathbf{v}^{temporal} \in \mathbb{R}^{H \times W \times C \times T}$, based on the spatial and temporal information of the satellite image sequence $\mathbf{I}$, the topography, vegetation, fuel maps $\mathbf{P}, \mathbf{G}, \mathbf{F}$, and the weather $\mathbf{W}$ as:

$$\mathbf{q} = SwinEnc(\mathbf{S}),$$
$$\mathbf{k}^{spatial} = SwinEnc([\mathbf{P}, \mathbf{G}, \mathbf{F}, \mathbf{I}]), \ \mathbf{v}^{spatial} = SwinEnc([\mathbf{P}, \mathbf{G}, \mathbf{F}, \mathbf{I}]),$$
$$\mathbf{k}^{temporal} = [SwinEnc(\mathbf{I}), MHA(W)], \ \mathbf{v}^{temporal} = [SwinEnc(\mathbf{I}), MHA(W)],$$
$$\mathbf{A} = softmax(\frac{\mathbf{q} \cdot \mathbf{k}^{spatial}}{\sqrt{W \times H}}) \cdot \mathbf{v}^{spatial} + softmax(\frac{\mathbf{q} \cdot \mathbf{k}^{temporal}}{\sqrt{T}}) \cdot \mathbf{v}^{temporal}, \qquad (1)$$

where $[\cdot]$, $SwinEnc$, $MHA$ and $softmax$ denote the feature concatenation, the encoder of Swin Transformer [56], multi-head attention, and softmax function.

Table 2: Comparison between S2R-FireTr and other AI models for fire forecast and backtracking.

| Method | Forecast | | | | | | Backtrack | | | | | |
| | Simulation Data | | | Real-world Data | | | Simulation Data | | | Real-world Data | | |
| | AUPRC | F1 | IOU | AUPRC | F1 | IOU | AUPRC | F1 | IOU | AUPRC | F1 | IOU |
|---|---|---|---|---|---|---|---|---|---|---|---|---|
| ConvLSTM [57] | 36.2 | 44.3 | 28.1 | 29.3 | 22.1 | 20.1 | 24.7 | 39.0 | 23.6 | 16.4 | 15.9 | 14.3 |
| Mau [58] | 59.4 | 67.4 | 50.2 | 43.6 | 49.8 | 41.9 | 54.7 | 60.2 | 35.3 | 40.1 | 45.5 | 32.6 |
| PredRNN-v2 [59] | 75.2 | 71.0 | 55.2 | 66.2 | 58.0 | 49.3 | 59.9 | 62.3 | 46.4 | 50.9 | 51.7 | 40.8 |
| Rainformer [60] | 79.7 | 78.8 | 69.6 | 67.2 | 65.5 | 52.0 | 73.3 | 71.9 | 57.0 | 54.6 | 52.4 | 42.9 |
| Earthformer [61] | 77.2 | 73.5 | 59.7 | 65.4 | 61.7 | 50.1 | 71.4 | 63.0 | 48.1 | 53.4 | 51.3 | 41.7 |
| SwinLSTM [62] | 77.1 | 71.2 | 56.2 | 62.5 | 60.3 | 48.9 | 72.5 | 65.3 | 52.4 | 53.8 | 49.5 | 40.3 |
| Earthfarsser [63] | 73.4 | 70.6 | 63.5 | 62.4 | 60.5 | 49.2 | 69.3 | 68.5 | 50.9 | 51.6 | 49.4 | 37.5 |
| ML-BPM [64] | 65.2 | 61.9 | 50.2 | 53.1 | 51.4 | 43.1 | 59.7 | 57.0 | 47.3 | 46.7 | 43.5 | 36.9 |
| OLDM [65] | 66.8 | 63.8 | 51.3 | 55.0 | 52.8 | 44.2 | 60.1 | 58.4 | 48.7 | 48.3 | 45.6 | 37.2 |
| **S2R-FireTr** | **87.3** | **83.2** | **71.2** | **72.9** | **69.6** | **56.4** | **78.6** | **73.5** | **58.1** | **63.9** | **60.3** | **46.9** |

The above dual cross-attention comprehensively constructs the correlation between fire areas and multi-modal spatial-temporal data. During network training, despite the Sim2Real gap between fire areas of the simulation and real situations, the dual cross-attention can still rely on the correlation between multi-modal data to learn fire area representations that are more consistent with the natural environment. It thereby reduces the negative impact of the Sim2Real gap on network training.

**Time-wise Fire Area Regression** Based on the source area representation $\mathbf{A}$ of the source sequence $\mathbf{S}$, we use the second module of S2R-FireTr (Figure 3(b)) to compute the target area representation $\mathbf{R} \in \mathbb{R}^{H \times W \times C \times T}$ of the future/history fire areas. Based on the target area representation $\mathbf{R}$, we regress the target sequence $\mathbf{T} \in \mathbb{R}^{H \times W \times T}$ of the forest fire areas in the forecast/backtracking task. We implement the area regression by a cross-attention. This module regards a set of timestamps $[p(1), ..., p(T)]$ as the query, the source area representation $\mathbf{A}$ as the key and value as:

$$\mathbf{q} = pos([p(1), ..., p(T)]), \ \ \mathbf{k} = conv(\mathbf{A}), \ \ \mathbf{v} = conv(\mathbf{A}), \ \ \mathbf{R} = softmax(\frac{\mathbf{q} \cdot \mathbf{k}}{\sqrt{W \times H}}) \cdot \mathbf{v}, \quad (2)$$

where $\mathbf{q}, \mathbf{k}, \mathbf{v} \in \mathbb{R}^{H \times W \times C \times T}$ represent the query, key, and value vectors. $pos$ means the positional encoding. We remark that the timestamps $[p(1), ..., p(t), ..., p(T)]$ in Eq. (2), which are used for computing the query vector $\mathbf{q}$, are unnecessarily continuous. $p(t)$ is the timestamp of the $t^{th}$ frame. This allows the model to be tested on real-world data, which may be temporally incomplete.

Given the target area representation $\mathbf{R}$, we regress the target sequence of binary masks $\widehat{\mathbf{T}} \in \mathbb{R}^{H \times W \times T}$ as:

$$\widehat{\mathbf{T}} = SwinDec(\mathbf{R}), \ \ \mathcal{L} = \|\mathbf{T} - \widehat{\mathbf{T}}\|, \quad (3)$$

where $SwinDec$ means the decoder of Swin Transformer. During the model training phase, we minimize the difference between the regressed sequences $\widehat{\mathbf{T}}$ and the ground-truth sequences $\mathbf{T}$.

## 4 Experimental Results

We compare S2R-FireTr with the empirical, physical, and AI models on the Sim2Real spread forecast and backtracking tasks. We train all AI models on the simulation data and evaluate their performances on the simulation and real-world data. These simulators are based on empirical and physical models and work without a training process. They only rely on the multi-modal environmental information to predict the fire spread during the evaluation phase. We evaluate the performances of these models in terms of AUPRC (Area Under the Precision-Recall Curve), Intersection over Union (IOU), and F1-score. We report each metric in percentage (%).

Table 3: Comparison between S2R-FireTr and simulators on the forecast task. We report the results in terms of AUPRC.

| Simulator | Real-world Data |
|---|---|
| FARSITE [1] | 55.9 |
| WFDS [9] | 61.2 |
| WRF-SFIRE [12] | 63.0 |
| **S2R-FireTr** | **72.9** |

Table 4: Impact of modalities on AI models for forecast. We report the results in terms of AUPRC.

| Method | Simulation Data | | | | | | Real-world Data | | | | | |
|---|---|---|---|---|---|---|---|---|---|---|---|---|
| | w/o Topo. | w/o Veg. | w/o Fuel | w/o Wea. | w/o Sat. | Full | w/o Topo. | w/o Veg. | w/o Fuel | w/o Wea. | w/o Sat. | Full |
| ConvLSTM [57] | 33.4 | 33.7 | 34.0 | 33.8 | 33.9 | 36.2 | 25.1 | 25.3 | 26.5 | 26.0 | 25.5 | 29.3 |
| Mau [58] | 56.2 | 56.7 | 57.1 | 56.5 | 56.8 | 59.4 | 38.5 | 38.7 | 40.3 | 38.4 | 39.0 | 43.6 |
| PredRNN-V2 [59] | 71.5 | 72.1 | 72.4 | 72.0 | 71.9 | 75.2 | 60.4 | 61.7 | 62.0 | 61.3 | 60.5 | 66.2 |
| Rainformer [60] | 77.1 | 77.3 | 77.7 | 77.2 | 77.5 | 79.7 | 62.7 | 63.0 | 64.8 | 62.7 | 62.9 | 67.2 |
| Earthformer [61] | 73.3 | 74.1 | 74.5 | 73.4 | 73.2 | 77.2 | 60.5 | 61.8 | 62.3 | 60.4 | 60.1 | 65.4 |
| SwinLSTM [62] | 73.5 | 74.8 | 74.9 | 74.0 | 73.9 | 77.1 | 59.0 | 59.1 | 60.3 | 60.1 | 59.5 | 62.5 |
| Earthfarsser [63] | 69.3 | 70.3 | 71.1 | 70.4 | 70.8 | 73.4 | 58.3 | 58.5 | 60.1 | 59.0 | 59.1 | 62.4 |
| ML-BPM [64] | 60.5 | 61.3 | 63.0 | 61.5 | 62.4 | 65.2 | 50.4 | 50.6 | 51.3 | 50.9 | 51.0 | 53.1 |
| OLDM [65] | 61.7 | 62.1 | 63.3 | 62.2 | 63.0 | 66.8 | 51.7 | 52.0 | 53.1 | 51.8 | 51.9 | 55.0 |
| **S2R-FireTr** | **82.1** | **83.1** | **85.2** | **83.0** | **83.5** | **87.3** | **66.9** | **67.7** | **69.3** | **67.0** | **67.4** | **72.9** |

## 4.1 Performances of Various Models on Forecast and Backtracking Tasks

We compare S2R-FireTr with the commonly used AI models that can be easily adapted to the forecast and backtracking tasks in Table 2. We train the AI models on the simulation data and test them on the simulation and real-world data of Sim2Real-Fire. S2R-FireTr better constructs the correlation between various modalities of data, thus achieving better performance than other methods.

In Table 3, we evaluate the simulators based on various empirical and physical models on the forecast task. All simulators are assessed on the same real-world data set and given the same environmental information for a fair comparison. S2R-FireTr remarkably outperforms all of the compared simulators on the real-world data. This is because S2R-FireTr employs time-wise fire area regression to forecast better fire areas based on temporally incomplete data in the real world. We provide the forecast and backtracking results in Figure 4.

## 4.2 Performances of Various Strategies for Using Multi-modal Data

In Tables 4 and 6, we evaluate different modalities of environmental information, including topography (Topo.), vegetation (Veg.), fuel (Fuel) maps, weather (Wea.), and satellite image sequence (Sat.), on AI models in the forecast and backtracking tasks. This excludes each data modality from the model training and testing. Compared to the models with the complete set of data modalities (Full), the absence of any modality leads to the performance degra-

Table 5: Impact of modalities on simulators for forecast. We report the results in terms of AUPRC.

| Method | Real-world Data | | | | | |
|---|---|---|---|---|---|---|
| | w/o Topo. | w/o Veg. | w/o Fuel | w/o Wea. | w/o Sat. | Full |
| FARSITE [1] | 50.7 | 51.3 | - | 49.2 | - | 55.9 |
| WFDS [9] | 58.4 | 52.1 | - | 59.7 | - | 61.2 |
| WRF-SFIRE [12] | 59.5 | 55.6 | - | 56.8 | - | 63.0 |
| **S2R-FireTr** | **66.9** | **67.7** | **69.3** | **67.0** | **67.4** | **72.9** |

dation of all AI models, demonstrating the importance of each modality. We find that S2R-FireTr outperforms other AI models in every case where a data modality is eliminated. It shows the robustness of S2R-FireTr.

We also evaluate the impact of each data modality on the simulators (i.e., FARSITE [1], WFDS [9], and WRF-SFIRE [12]) in the forecast task (see Table 5). Compared to all modalities utilized in Tables 4, 5 and 6, the absence of each data modality results in a performance degradation of 2∼10%. It means that all data provided in the Sim2Real-Fire dataset helps analyze wildfires. We remark that the fuel map is the prerequisite for starting all simulators. Thus, we omit the comparison between the simulations without the fuel map.

## 4.3 Ablation Study of S2R-FireTr

We remove one or more critical components of S2R-FireTr (i.e., Environment-guided Area Representation Learning and Time-wise Fire Area Regression) to study their effectiveness on the forecast and backtracking tasks in Table 7. Compared to the entire model of S2R-FireTr, the alternatives without

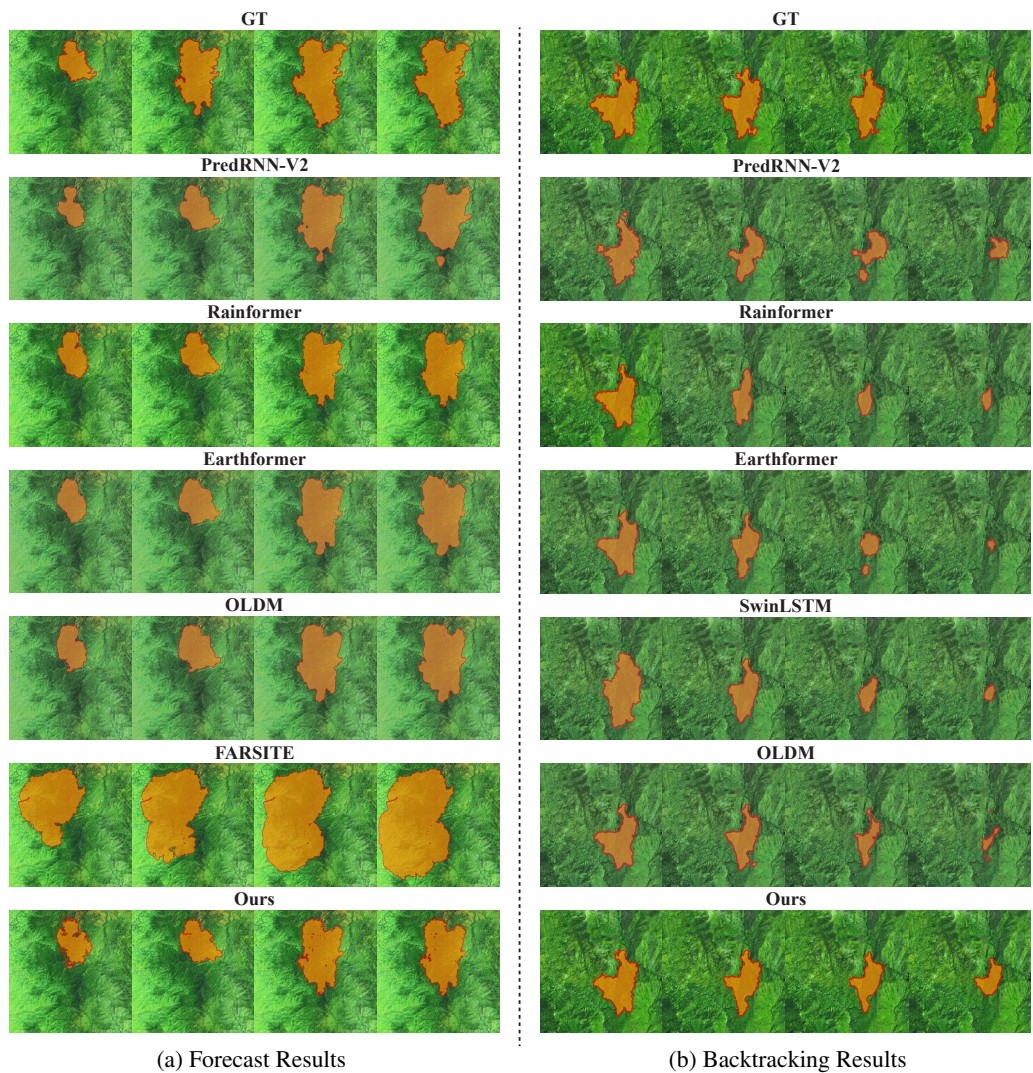

(a) Forecast Results          (b) Backtracking Results

Figure 4: Forecast and backtracking results of different methods.

Table 6: Impact of modalities on AI backtracking models. We report the results in terms of AUPRC.

| Method | Simulation Data | | | | | | Real-world Data | | | | | |
|---|---|---|---|---|---|---|---|---|---|---|---|---|
| | w/o Topo. | w/o Veg. | w/o Fuel | w/o Wea. | w/o Sat. | Full | w/o Topo. | w/o Veg. | w/o Fuel | w/o Wea. | w/o Sat. | Full |
| ConvLSTM [57] | 20.3 | 21.0 | 21.4 | 21.3 | 20.8 | 24.7 | 13.3 | 13.5 | 14.0 | 13.6 | 13.5 | 16.4 |
| Mau [58] | 50.8 | 51.2 | 51.3 | 50.4 | 50.7 | 54.7 | 35.7 | 35.8 | 36.1 | 35.4 | 35.7 | 40.1 |
| PredRNN-V2 [59] | 54.6 | 54.3 | 55.9 | 55.0 | 54.8 | 59.9 | 45.3 | 45.2 | 46.1 | 45.7 | 45.6 | 50.9 |
| Rainformer [60] | 70.7 | 71.2 | 71.4 | 71.0 | 70.9 | 73.3 | 50.1 | 50.7 | 51.6 | 50.4 | 50.5 | 54.6 |
| Earthformer [61] | 66.4 | 67.3 | 67.8 | 66.1 | 67.0 | 71.4 | 48.7 | 49.0 | 49.3 | 47.7 | 47.8 | 53.4 |
| SwinLSTM [62] | 67.8 | 68.1 | 68.4 | 67.9 | 67.4 | 72.5 | 48.3 | 49.2 | 49.4 | 48.3 | 47.9 | 53.8 |
| Earthfarsser [63] | 64.5 | 65.0 | 65.4 | 63.3 | 63.7 | 69.3 | 46.8 | 47.0 | 47.2 | 46.8 | 46.0 | 51.6 |
| **S2R-FireTr** | **75.7** | **75.8** | **76.1** | **74.9** | **75.0** | **78.6** | **59.8** | **60.0** | **61.3** | **59.4** | **60.7** | **63.9** |

these components, designed for learning correlation between multiple modalities of environmental information from the temporally incomplete data, remarkably degrade the performances.

In Table 8, we further study the impact of the input and output length of the temporal data (i.e., weather and satellite images) on the performance of S2R-FireTr. Excessively long input and output sequences degrade the performances. It demonstrates that forecasting and backtracking the long-term wildfire areas are highly challenging tasks. On the other hand, we find that the performance of

Table 7: Ablation study of key components on the forecast and backtracking tasks. EARL and TFAR mean environment-guided area representation learning and time-wise fire area regression.

| EARL | TFAR | Forecast | | | | | | Backtrack | | | | | |
|---|---|---|---|---|---|---|---|---|---|---|---|---|---|
| | | Simulation Data | | | Real-world Data | | | Simulation Data | | | Real-world Data | | |
| | | AUPRC | F1 | IOU | AUPRC | F1 | IOU | AUPRC | F1 | IOU | AUPRC | F1 | IOU |
| ✗ | ✗ | 70.1 | 74.5 | 59.8 | 60.2 | 64.8 | 45.3 | 58.7 | 63.4 | 46.4 | 42.4 | 50.2 | 33.8 |
| ✗ | ✓ | 79.0 | 82.2 | 70.1 | 63.9 | 65.8 | 47.7 | 75.2 | 71.9 | 56.3 | 58.5 | 54.5 | 37.7 |
| ✓ | ✗ | 83.0 | 82.5 | 70.3 | 70.1 | 66.7 | 50.1 | 76.1 | 72.0 | 56.5 | 59.0 | 57.1 | 39.7 |
| ✓ | ✓ | **87.3** | **83.2** | **71.2** | **72.9** | **69.6** | **56.4** | **78.6** | **73.5** | **58.1** | **63.9** | **60.3** | **46.9** |

Table 8: Impact of sequence length on the forecast and backtracking tasks.

| Squence length | Forecast | | | | | | Backtrack | | | | | |
|---|---|---|---|---|---|---|---|---|---|---|---|---|
| | Simulation Data | | | Real-world Data | | | Simulation Data | | | Real-world Data | | |
| | AUPRC | F1 | IOU | AUPRC | F1 | IOU | AUPRC | F1 | IOU | AUPRC | F1 | IOU |
| 1 | 83.3 | 77.4 | 65.1 | 68.4 | 64.7 | 55.4 | 70.2 | 67.3 | 50.6 | 55.9 | 50.2 | 36.6 |
| 2 | 85.0 | 80.4 | 67.3 | 70.1 | 67.0 | 55.9 | 75.1 | 69.2 | 52.9 | 59.7 | 55.6 | 38.5 |
| 3 | 87.3 | 83.2 | 71.2 | **72.9** | **69.6** | **56.4** | **78.6** | **73.5** | **58.1** | **63.9** | **60.3** | **46.9** |
| 4 | **88.2** | **84.3** | **72.9** | 70.6 | 66.9 | 50.3 | 78.2 | 71.4 | 56.2 | 59.9 | 50.1 | 40.9 |
| 5 | 86.6 | 83.5 | 71.7 | 67.6 | 59.3 | 42.1 | 76.2 | 69.2 | 52.9 | 49.1 | 45.1 | 30.8 |
| 6 | 85.6 | 82.7 | 70.0 | 63.3 | 54.1 | 38.7 | 73.4 | 68.0 | 50.6 | 42.6 | 40.3 | 27.5 |

S2R-FireTr within six frames is satisfactory. Given that the sequence length of fire areas in real-world applications is relatively short, we consider the practicability of S2R-FireTr to be solid.

## 5 Conclusion

This paper introduces the Sim2Real-Fire dataset with 1M simulated scenarios and 1K realistic wildfire scenarios for training and testing AI models that forecast and backtrack wildfires in the real world. This dataset is meaningful for the Sim2Real investigation of wildfire forecast and backtracking. Technically, we contribute a deep transformer, S2R-FireTr, trained on the simulated scenarios. S2R-FireTr surpasses state-of-the-art methods, demonstrating the potential of minimizing the Sim2Real gap between the simulated and realistic wildfire scenarios. The sim2Real-Fire dataset is limited as it only includes wildfire scenarios from certain countries and periods due to the limited budget for data acquisition in reality. This closed nature reduces the richness of the training data, limiting the model's ability to generalize to unknown environmental conditions. To address this, we advocate for dynamic data acquisition methods to transform the dataset into an open resource. In the future, we will extend our dataset and method to a broader range of wildfire analysis tasks, where we need to transfer the fire spread patterns learned from the simulated scenarios to the real world. People can access our dataset, a video detailing the dataset creation process, relevant documentation, and model code via `https://github.com/TJU-IDVLab/Sim2Real-Fire`.

## 6 Broader Impacts

This paper has several potential positive societal impacts: the proposed Sim2Real-Fire dataset, a multi-modal dataset with temporal data, is designed to facilitate deep learning models on the relevant tasks for wildfire analysis. The proposed S2R-FireTr model provides a comprehensive understanding of the multiple environmental factors influencing forecast and backtracking, thereby enhancing the accuracy of wildfire prediction. This work has inconspicuous negative societal impacts.

## 7 Acknowledgement

The Key Science and Technology Program of the Ministry of Emergency Management of the People's Republic of China (2024EMST010102) fully supported this work.

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
