# Sim2Real-Fire: A Multi-modal Simulation Dataset for Forecast and Backtracking of Real-world Forest Fire –Supplementary material–

This supplementary material provides the following information:

- Section A: Supplementary Details of Sim2Real-Fire;
- Section B: Implementation Details of S2R-FireTr;
- Section C: Limitation Analysis on Temporal Interval;

People can access our dataset, a video detailing the dataset creation process, relevant documentation, and model code via `https://github.com/TJU-IDVLab/Sim2Real-Fire`. These resources are available under the Apache-2.0 license.

## A    Supplementary Details of Sim2Real-Fire

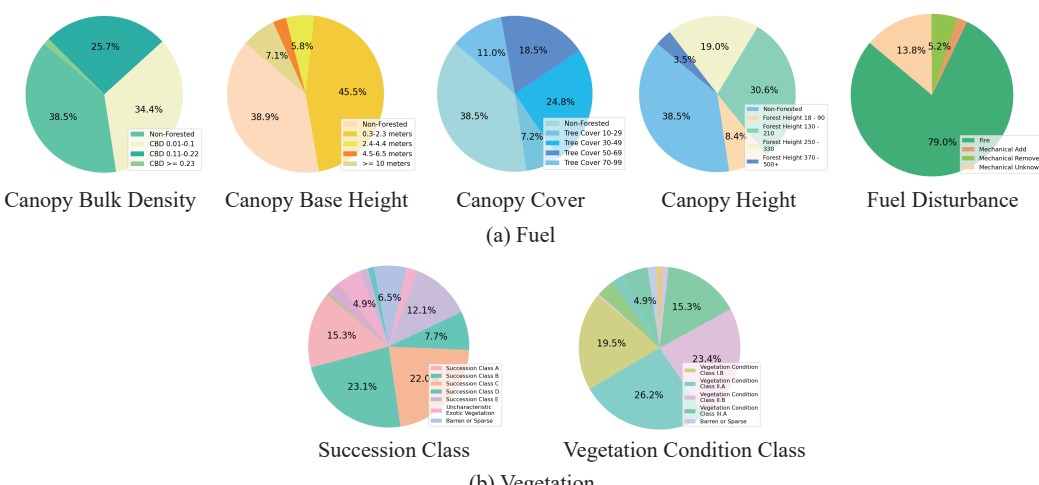

Figure 1: Details of the distributions of (a) fuel and (b) vegetation.

In Figure 1, we provide more details of the distribution of fuel and vegetation, which are only briefly presented in the main paper due to the limited pages. In Figures 2 and 3, we provide examples of simulated and real satellite image sequences of wildfire. **For a better visualization, we align the annotated masks of fire areas to the clear satellite images without smoke, fog, cloud, etc. The examples of satellite images with the real fire areas can be found in the anonymous github repository**.

## B    Implementation Details of S2R-FireTr

We implement the S2R-FireTr model for wildfire forecasting and backtracking using PyTorch. We train the model on an NVIDIA 3090 GPU with a learning rate of 0.0001 using the Adam optimizer. We set the batch size to 4. Each training sequence contains six frames; each resized to $256{\times}256$ pixels. We train S2R-FireTr for ten epochs.

38th Conference on Neural Information Processing Systems (NeurIPS 2024) Track on Datasets and Benchmarks.

Soberanes

Augustcomplex

Beachibreek

Castle

Lionshead

Pinegulch

Windy

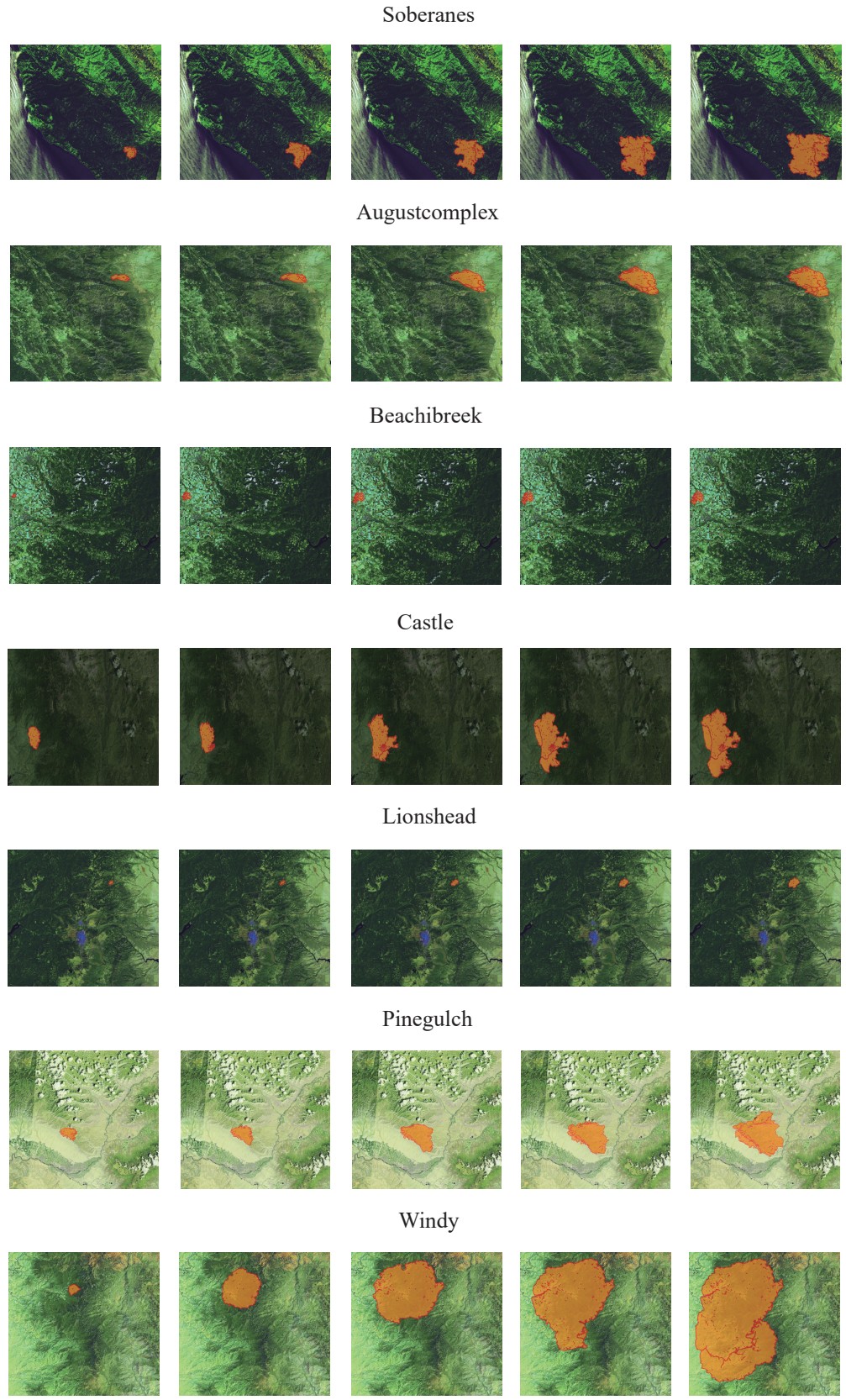

Figure 2: Examples of simulated satellite image sequences of wildfire.

Soberanes

Augustcomplex

Beachibreek

Castle

Lionshead

Pinegulch

Windy

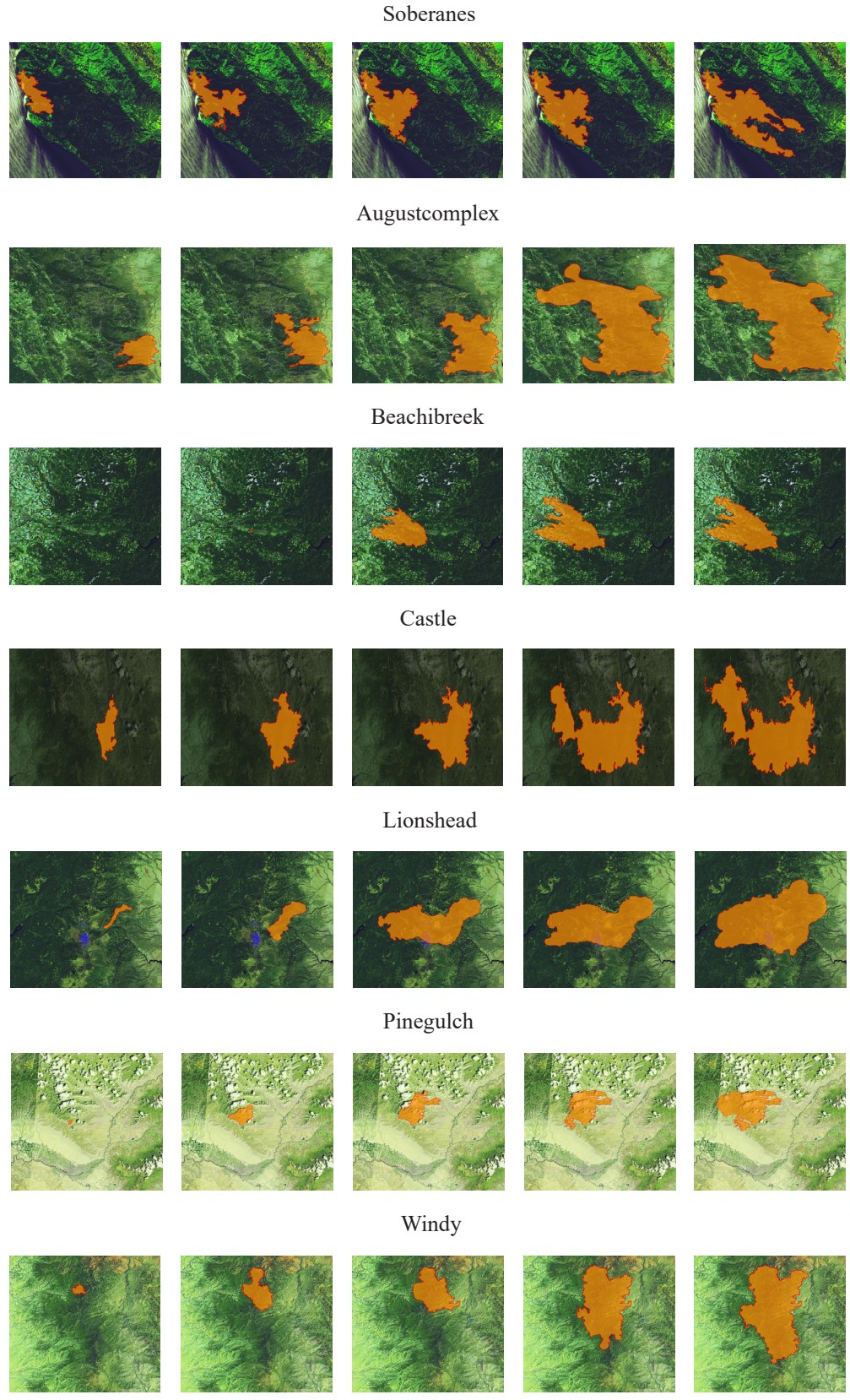

Figure 3: Examples of real satellite image sequences of wildfire.

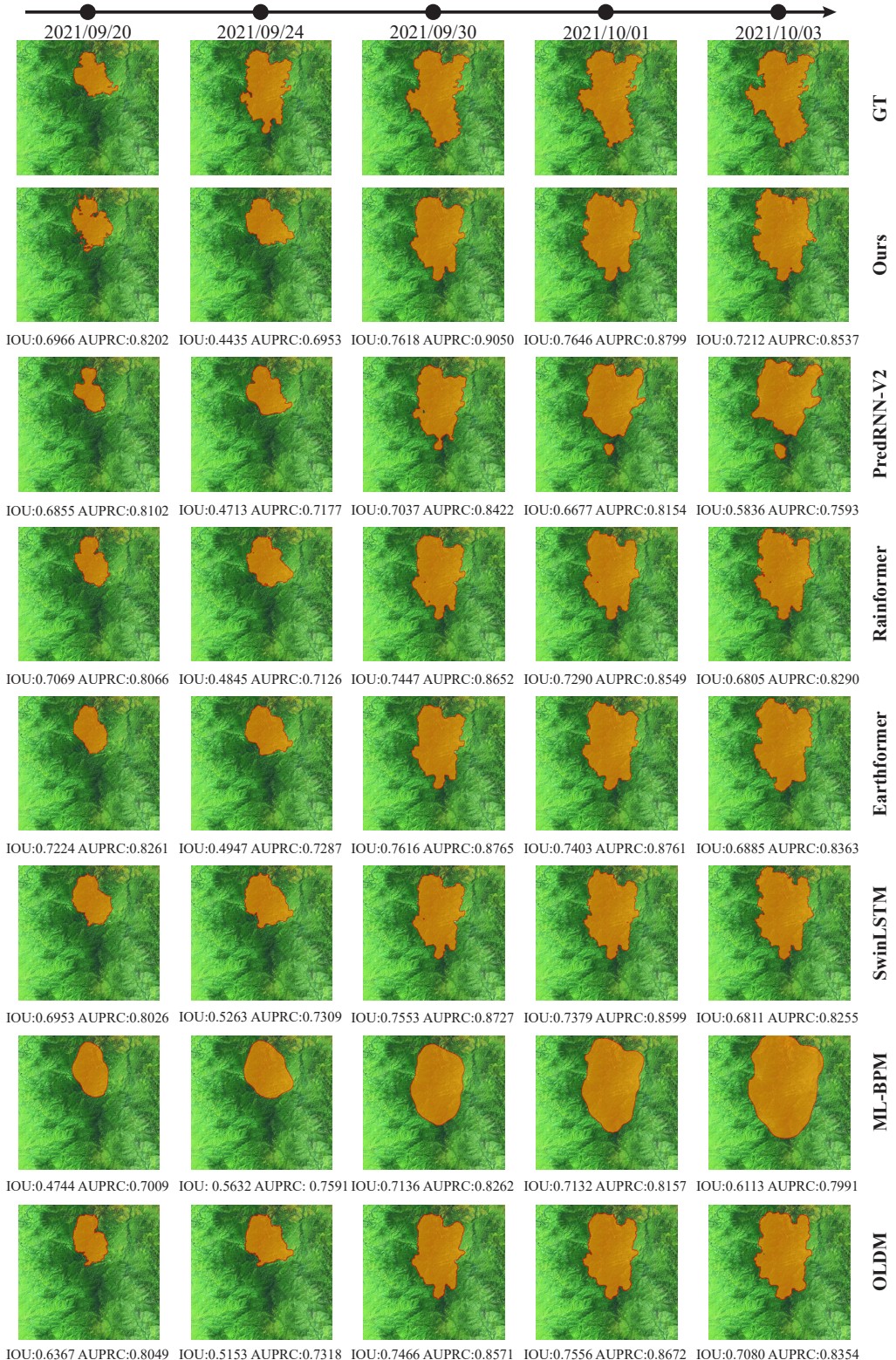

Figure 4: Results of the forecast task.

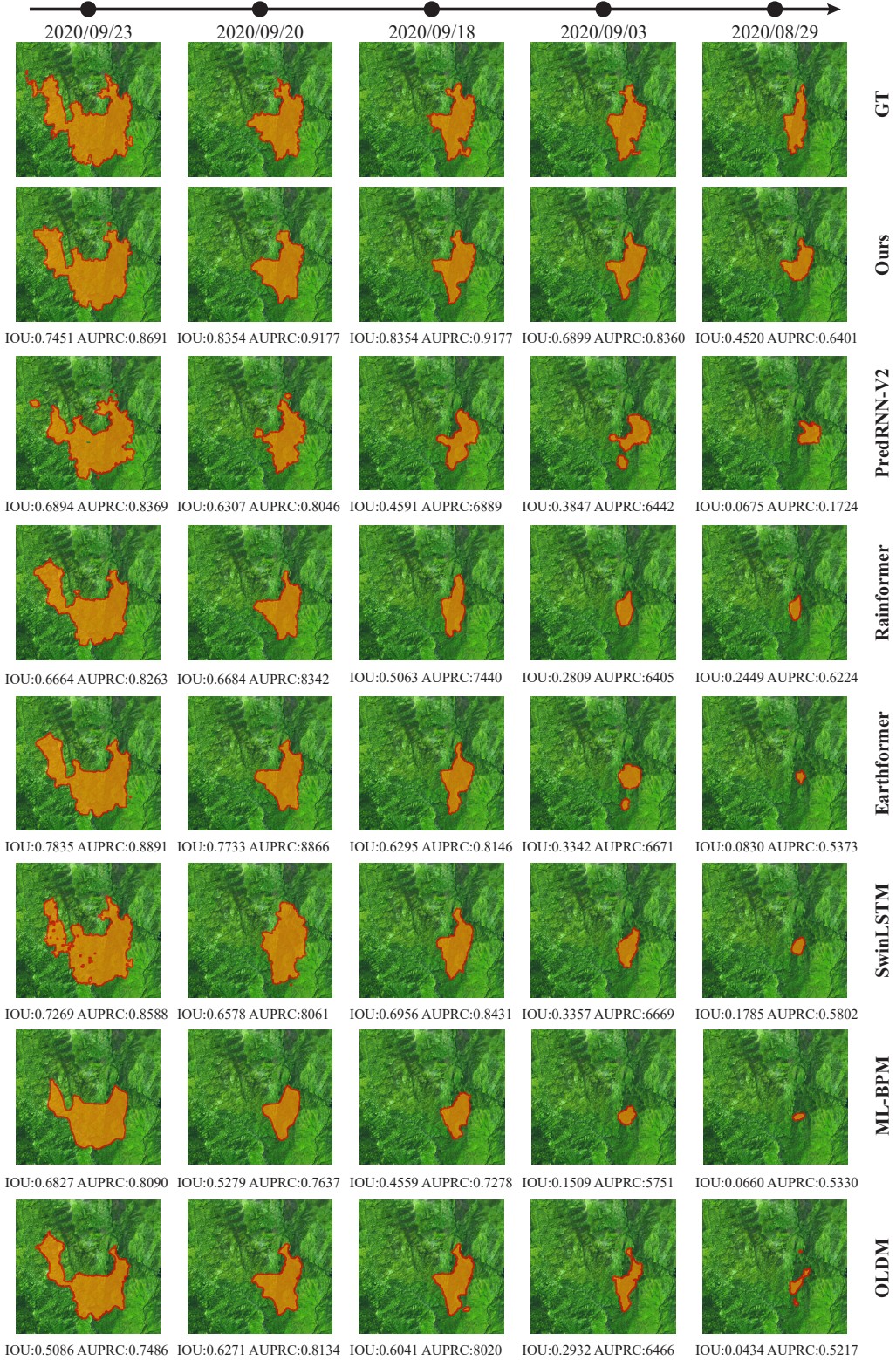

Figure 5: Results of the backtracking task.

## C   Limitation Analysis on Temporal Interval

In Table 1, we present the performance of S2R-FireTr, which is trained with different temporal intervals. By increasing the temporal interval between frames of the simulated wildfire scenarios during model training (see interval of 5∼20), S2R-FireTr shows improvements in forecasting and backtracking real wildfire scenarios. This improvement is primarily because real-world wildfire frames are captured by satellites orbiting the Earth. Satellite orbits around the Earth typically involve relatively large temporal intervals, and this aligns with the training data. By randomly setting the interval (maximum of 20), we better match the training data to real-world scenarios, where intervals between frames are usually uneven (see the last row). However, using excessively large intervals inevitably degrades S2R-FireTr's performance (see interval of 40∼80). Large intervals reduce the available training data, leading to a decline in S2R-FireTr's performance. In Figure 6, we illustrate several instances of failure. This underscores the need to enhance S2R-FireTr's capacity to manage changes in fire areas when dealing with excessively large intervals.

Table 1: Limitation analysis on temporal interval.

| Interval(h) | Forecast | | | Backtrack | | |
|---|---|---|---|---|---|---|
| | AUPRC | F1 | IOU | AUPRC | F1 | IOU |
| 5 | 54.7 | 52.4 | 46.3 | 43.8 | 45.6 | 38.9 |
| 10 | 56.2 | 53.1 | 48.8 | 45.7 | 47.3 | 39.1 |
| 15 | 57.0 | 64.9 | 50.0 | 47.8 | 50.1 | 41.6 |
| 20 | 58.7 | 67.0 | 52.5 | 49.2 | 52.4 | 43.3 |
| 40 | 55.1 | 62.3 | 49.6 | 45.3 | 46.1 | 39.5 |
| 80 | 45.1 | 50.7 | 40.1 | 40.0 | 41.2 | 35.4 |
| **Random** | **72.9** | **69.6** | **56.4** | **63.9** | **60.3** | **46.9** |

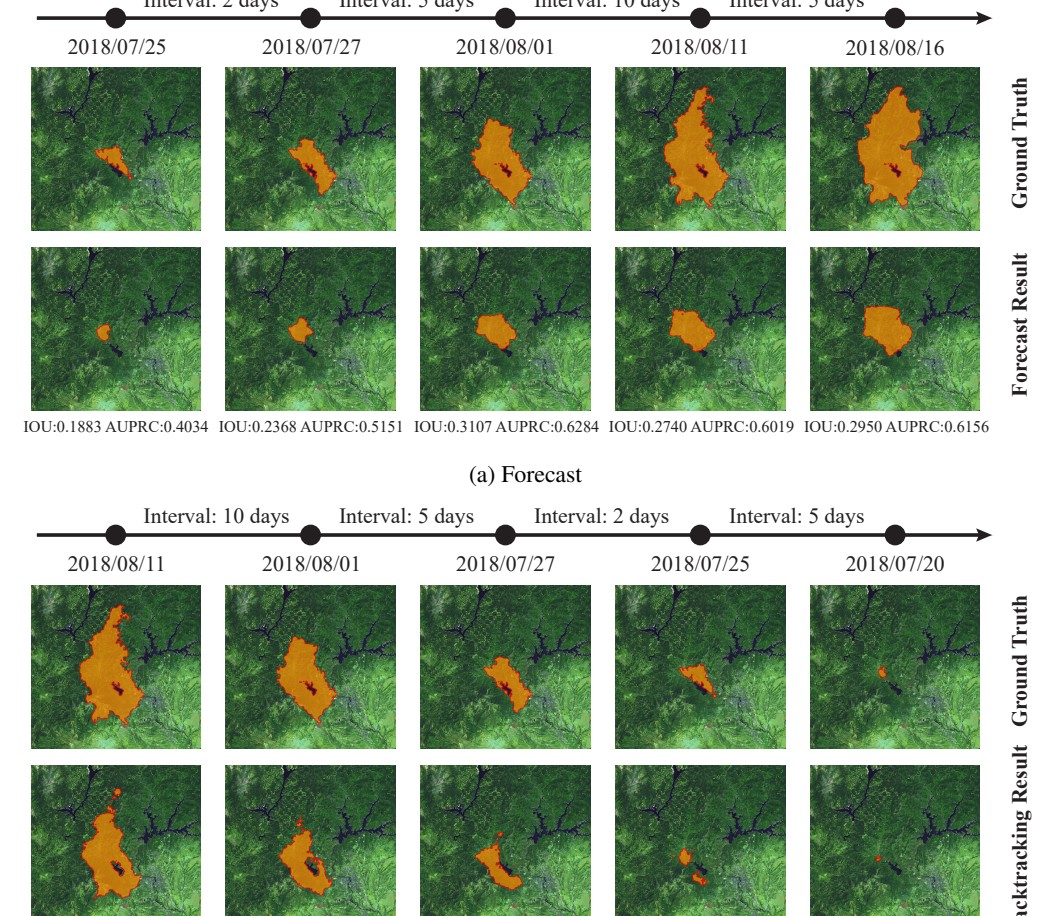

(a) Forecast

(b) Backtrack

Figure 6: Failure cases of (a) forecast and (b) backtracking.