# OpenReview forum: "Sim2Real-Fire: A Multi-modal Simulation Dataset for Forecast and Backtracking of Real-world Forest Fire"
_NeurIPS.cc/2024/Datasets_and_Benchmarks_Track — NeurIPS 2024 Track Datasets and Benchmarks Poster_

### Official Review · Reviewer_JYQM · 2024-07-23
**Sim2Real-Fire: A Multi-modal Simulation Dataset for Forecast and Backtracking of Real-world Forest Fire**

**Rating:** 6
**Confidence:** 3

**Review:**

The paper tackles the challenge of simulating realistic wildfire scenarios and bridging the Sim2Real gap, which is critical for effective wildfire prediction and management. While the approach is innovative and the dataset comprehensive, there are several significant weaknesses and areas that require improvement:

Quality:

1. Comparative Analysis with Existing Datasets: The paper fails to provide a thorough comparison with existing similar datasets such as MODIS Thermal Anomaly, VIIRS Thermal Anomaly, GOES Wildfire, NOAA HMS, NIFC Wildfire Perimeters, WildfireSpreadTS, and Mesogeos. The novelty and advantages of Sim2Real-Fire over these established datasets are not clearly articulated. Specific metrics and detailed evaluations are missing.

2. Dataset Characteristics and Quality: The paper mentions the inclusion of 1 million simulated scenarios, but there is limited discussion on the quality, variability, and fidelity of these simulations. The potential biases introduced by the simulators (FARSITE, WFDS, WRF-SFIRE) and how they compare to real-world fire behavior are not critically examined.

Clarity:

1. Explanation of Data Modalities: While the paper outlines the various data modalities (topography, vegetation, fuel, weather, and satellite images), it lacks detailed descriptions and justifications for their selection and integration. The preprocessing steps and any challenges encountered are not adequately discussed.

2. Model Architecture Details: The description of the S2R-FireTr model is somewhat opaque. The paper should provide a clearer breakdown of the model components, training procedures, and hyperparameter settings. Additionally, there is insufficient explanation of how the model handles temporally incomplete data.

Originality:

1. Innovation in Model Design: The use of transformers for wildfire prediction is novel; however, the paper does not sufficiently differentiate this approach from existing transformer-based models in other domains. The unique contributions and adaptations specific to wildfire prediction are not well-highlighted.

2. Real-World Impact: The practical implications and potential real-world applications of the model and dataset are underexplored. The paper should discuss how the model’s predictions can be integrated into existing wildfire management systems and what benefits it offers over current practices.

Significance:

1. Evaluation Metrics and Benchmarks: The evaluation focuses on standard metrics like AUPRC, F1, and IOU. However, these metrics might not fully capture the effectiveness of wildfire spread prediction in practical scenarios. Including additional metrics relevant to wildfire management, such as false negatives and prediction lead time, would strengthen the evaluation.

2. Generalization to Other Regions: The dataset primarily focuses on the United States, Canada, and Mexico. The paper should discuss the potential limitations and challenges in applying the model to other geographic regions with different environmental conditions and fire behavior patterns.

**Strengths:**

1. Comprehensive Dataset: The Sim2Real-Fire dataset is extensive, including multiple data modalities and a significant number of scenarios, which can be valuable for training robust AI models.

2. Innovative Model: The proposed S2R-FireTr model demonstrates strong performance in both simulated and real-world scenarios, showcasing the potential of transformer-based architectures for wildfire prediction.

**Additional Feedback:**

The paper has potential but requires significant improvements in terms of detailed comparisons with existing work, comprehensive dataset descriptions, transparent model explanations, and practical implications. Addressing these issues will enhance the paper's contribution to the field and its potential impact on wildfire prediction and management.

**Clarity:**

The paper is generally well-written but lacks clarity in key areas such as dataset description, model architecture, and evaluation methods. Improving these sections will significantly enhance the paper’s readability and comprehensibility.

**Correctness:**

The claims made in the submission appear to be correct, but they are not sufficiently supported by detailed evaluations and comparisons with existing methods and datasets. More rigorous validation and benchmarking are required.

**Documentation:**

The dataset documentation is insufficient. The paper should include detailed information on data collection, preprocessing, organization, availability, and maintenance. Ethical and responsible use considerations are also lacking.

**Ethics:**

No significant ethical concerns are identified, but the paper should discuss the responsible use of the dataset and potential societal impacts of deploying the model in real-world scenarios.

**Limitations:**

The paper acknowledges the limitations related to the Sim2Real gap and the potential inaccuracies in simulated data. However, it does not provide concrete solutions or mitigation strategies for these issues. Further discussion on the limitations and potential biases in the dataset and model is needed.

**Opportunities For Improvement:**

Opportunities For Improvement

1. Detailed Comparison with Existing Datasets: Provide a comprehensive analysis comparing Sim2Real-Fire with established datasets, highlighting the unique contributions and advantages.

2. Enhanced Dataset Description: Include more detailed descriptions of the data modalities, preprocessing steps, and quality assessments to give a clearer picture of the dataset's robustness and reliability.

3. Model Explanation and Validation: Offer a more transparent and detailed explanation of the S2R-FireTr model architecture, training procedures, and validation techniques. Include ablation studies to demonstrate the importance of each component.

4. Practical Implications: Discuss the real-world applications and potential impact of the model on wildfire management, including integration into existing systems and practical benefits.

5. Geographic Generalization: Address the applicability and challenges of using the dataset and model in different geographic regions with varied environmental conditions.

**Relation To Prior Work:**

The paper does not adequately discuss how it differs from previous contributions, particularly in comparison to established wildfire datasets and prediction models. A more thorough literature review and comparison are necessary.

**Summary And Contributions:**

The paper presents Sim2Real-Fire, a multi-modal dataset containing 1 million simulated and 1,000 real-world wildfire scenarios for training and evaluating AI models aimed at forecasting and backtracking wildfire spread. It introduces a novel deep transformer network, S2R-FireTr, which leverages this dataset to achieve superior performance compared to state-of-the-art methods.

---

> ### Author Rebuttal · Authors · 2024-08-16
>
> **(4/4)**
>
> **Q6: The practical implications and potential real-world applications of the model and dataset are underexplored. The paper should discuss how the model’s predictions can be integrated into existing wildfire management systems and what benefits it offers over current practices.**
>
> We envision that S2R-FireTr can be applied to the forecast and backtracking sub-systems of wildfire management:
>
> - The forecast function of S2R-FireTr is installed in the form of software onto the cloud computing platform of the fire management system or the computational server for on-site command. Upon receiving satellite data transmitted to the computer, S2R-FireTr can quickly locate current and future fire areas. These current and future fire areas can be further transmitted to helicopters and fire trucks equipped for water spraying and fire extinguishing, providing references for firefighters to assist them in spraying water toward the current fire areas and flame retardants towards potential future fire areas.
> - The backtracking function of S2R-FireTr can also be integrated into the fire management system to analyze historical fire data and fire accident investigation. Fire departments and disaster research institutions can input the historical fires into the backtracking model of S2R-FireTr to complete the fire areas in earlier history, thereby locating the initial ignition area. This helps people explain the causes of fires. For example, a location with abundant flammable vegetation and numerous visitors is more likely to be prone to human-caused fires. More comprehensive information can alert people to similar fire risks.
>
> **Q7: The evaluation focuses on standard metrics like AUPRC, F1, and IOU. However, these metrics might not fully capture the effectiveness of wildfire spread prediction in practical scenarios. Including additional metrics relevant to wildfire management, such as false negatives and prediction lead time, would strengthen the evaluation.**
>
> We respectfully point out that AUPRC, F1, and IOU are the most commonly used metrics to evaluate the accuracy of pixel-level image area recognition. The metric AUPRC considers true/false positive and false negative, which is more comprehensive than the metric of false negative alone. Its formulation is:
>
> $\text{AUPRC} = \int_{0}^{1} \text{Precision}(\text{Recall}) \~d(\text{Recall}),$
> where
> $\text{Precision} = \frac{\text{True Positives}}{\text{True Positives} + \text{False Positives}}$,
> $\text{Recall} = \frac{\text{True Positives}}{\text{True Positives} + \text{False Negatives}}$.
>
> We clarify that the prediction lead time is inapplicable to evaluate S2R-FireTr. As expected, S2R-FireTr can be integrated into the fire management system. The forecast and backtracking functions of S2R-FireTr indicate the future fire spread area and high-risk ignition areas while the fire department conducts fire prevention and extinguishing in reality. According to our understanding, the prediction lead time refers to the time difference between the forecast of the fire areas and the execution of fire extinguishing. Our research paper focuses more on the accuracy of forecasting the fire areas, while the actual execution of extinguishing is beyond our research scope and needs to be carefully considered in future works.
>
>
> **Q8: The dataset primarily focuses on the United States, Canada, and Mexico. The paper should discuss the potential limitations and challenges in applying the model to other geographic regions with different environmental conditions and fire behavior patterns.**
>
> We are very grateful for your advice. Suppose a data set can cover as much information as possible about the worldwide topography, vegetation, fuel, weather conditions, etc. In that case, its training data can effectively improve the generalization of fire area forecast and backtracking models. Based on this ideal intuition, we discuss the limitation and potential opportunity brought by our Sim2Real-Fire dataset:
>
> - **Limitation**. Although the Sim2Real-Fire dataset has unique attributes (see our response to **Q1**), it is a closed set like most existing datasets. Sim2Real-Fire only includes wildfire scenarios from certain countries and periods due to the limited budget for data acquisition in reality. It significantly reduces the richness of the training data. The model, trained on the data with a few known categories of topography, vegetation, fuel, and weather conditions, eventually lacks the generalization power to forecast the fire areas under unknown environmental conditions.
> - **Opportunity**. As we have stated in **Limitation**, it is inevitable that closed datasets cannot provide comprehensive environmental information. Therefore, a cost-effective and fast data acquisition method must be considered to dynamically add more data and change the closed set to the open dataset. Some existing datasets, like MODIS Thermal Anomaly, VIIRS Thermal Anomaly, GOES Wildfire, NOAA HMS, and NIFC Wildfire Perimeters, are open to refresh the environmental data. However, these datasets rely on continuous satellite monitoring, leading to high data acquisition costs. In addition, there is a large amount of redundant data without fire areas, thus posing difficulties for data cleaning and model training. This paper advocates minimizing the Sim2Real gap between the simulated and real-world scenarios, which is the core of our paper apart from the dataset and methodology contributions. **Minimizing the Sim2Real gap is challenging, but once we can address it, we can use cost-effective simulators to produce a large amount of data rapidly**. By combining different environmental conditions, we can produce large-scale data to add to the Sim2Real-Fire dataset, which becomes an open set for model training. This fashion lets the model witness simulated fire areas under different environmental conditions that probably appear in real-world scenarios, where the generalization power of the model is enhanced.

---

> ### Author Rebuttal · Authors · 2024-08-16
>
> **(3/4)**
>
> **Q3: The paper lacks detailed descriptions and justifications for their selection and integration. The preprocessing steps and any challenges encountered are not adequately discussed.**
>
> We plan to add details about data selection and integration from the following aspects:
>
> - **Wildfire localization**. We selected the fire data from 2013 to 2023 in the United States, Canada, and Mexico from the Global Wildfire Database and recorded the longitude and latitude of fire occurrences.
> - **Environmental information association**. Based on the latitudes and longitudes obtained from the Global Wildfire Database, we retrieved the corresponding topography, vegetation, and fuel information for each fire scenario from the Landfire website, as well as the corresponding weather information from the Remote Automatic Weather Station website and meteomanz.com.
> - **Production of simulated fire**. We input the environmental information into different simulators (i.e., FARSITE, WFDS, and WRF-SFIRE), producing the masks of the changing fire areas.
> - **Pre-processing and annotation of real-world fire**. We exclude satellite images that fail to clearly depict fire regions owing to the obstruction caused by dense clouds or smoke. Next, we upload the satellite data into the NV5 GEOSPATIAL platform, where we select the Radiometric Calibration tool from the comprehensive toolbox to procure a precisely calibrated radiance image. Following this, we employ the Quick Atmospheric Correction feature to acquire a FLAASH atmospheric correction image, ensuring utmost accuracy. To maintain consistency and capture the intricate landscapes of Earth at a resolute 30-meter spatial resolution, we utilize the Resample tool within the ArcGIS toolbox. The preprocessed satellite images are imported into ArcGIS for manual annotation. After three individuals simultaneously annotate a single satellite image following the aforementioned method, their output results are compared, and discrepancies in annotations are further verified. The final annotated results are then outputted.
>
> We have provided a video on the homepage of our GitHub repository to illustrate the above steps more directly.
>
>
> **Q4: The paper should provide a clearer breakdown of the model components, training procedures, and hyper-parameter settings. Additionally, there is insufficient explanation of how the model handles temporally incomplete data.**
>
> We respectfully clarify that Section 4.3 “Ablation Study of S2R-FireTr” of the paper has introduced the ablation study of the network components, which are **Environment-guided Area Representation Learning (EARL)** and **Time-wise Fire Area Regression (TFAR)** newly proposed in this paper. These components are tailor-made for Sim2real wildfire forecast and backtracking. We have also provided some training and hyper-parameter settings in Section C of the supplementary file. In Table 8 of the paper and Table 2 of the supplementary file, we study the impact of the critical hyper-parameters (i.e., the length of the sequential fire areas and the time interval between adjacent frames of fire areas) on the model training. Please also note that the entire code package with all details of hyper-parameters, training, and testing has been released via this anonymous GitHub repository https://github.com/anonymous-github-Sim2Real-Fire/Sim2Real-Fire, for a more transparent examination and reproduction. We have made every effort to provide a significant amount of information in the paper and supplementary file. We would be pleased to provide additional details if the reviewer requests more.
>
>
> **Q5: The use of transformers for wildfire prediction is novel; however, the paper does not sufficiently differentiate this approach from existing transformer-based models in other domains. The unique contributions and adaptations specific to wildfire prediction are not well-highlighted.**
>
> Thanks for your suggestion. Apart from the unique properties of our Sim2Real-Fire data described in our response to your **Q1**, our transformer network S2R-FireTr also has unique contributions from two aspects:
>
> - S2R-FireTr has the transformer-based component of **EARL**. During the model training on the simulation data, this component uses cross attention to take the simulated fire areas as the query. Environmental information (i.e., topography, vegetation, fuel, and weather) plays the key and value. **EARL** selects consistent information from the simulated fire areas and multiple modalities of environmental factors to forecast/backtrack fire areas. The consistent information generally appears in reality, thus allowing the model training on the simulation data to leverage more realistic information. **Typically, the general deep networks mix the simulated fire areas and the environmental information as input without considering the information selection like S2R-FireTr**.
> - S2R-FireTr has another transformer-based component of **TFAR**, which can forecast and backtrack fire areas at arbitrary moments. This is done by adding a special design of timestamps embedding into the cross attention. These timestamps have unequal temporal intervals. Specifically, we embed these timestamps (see Section A of the supplementary file) into the latent representations input into the cross-attention with the temporal and spatial representations from **EARL**. This procedure is illustrated in Figure 3. This is how the model handles the temporal incomplete data. It allows TFAR to better consider the temporal incompletion of fire areas, which may be captured by different satellites flying over at unequal time intervals. **For more general purposes, the deep networks assume the time intervals between sequential frames are equal, thus lagging behind S2R-FireTr.**
>
> Tables 2, 4, and 6 of the paper and Table 1 of the supplementary file compare S2R-FireTr with the general deep networks like CNN, LSTM, and Transformer, where S2R-FireTr achieves better results.

---

> ### Author Rebuttal · Authors · 2024-08-16
>
> **(2/4)**
>
> **Q2: There is limited discussion on the quality, variability, and fidelity of these simulations. The potential biases introduced by the simulators (FARSITE, WFDS, WRF-SFIRE) and how they compare to real-world fire behavior are not critically examined.**
>
> **#1. Clarification on Our Contribution:**
>
> We agree that building a simulated dataset with high quality, variability, and fidelity levels but without bias is critical for training wildfire recognition models. As we have discussed in lines 55-64 of the paper, the quality and bias of the simulated data are difficult to measure in reality. Rather than using tricky methods to work around the issues of low quality and bias in the dataset, this paper makes a step to push the limit of using the simulated data with potentially low-quality and biased information for training. To reduce the impact of low-quality and biased information for model training, our Sim2Real dataset contains variant data produced by various simulators.
>
> Furthermore, our S2R-FireTr model has a transformer-based component of **Environment-guided Area Representation Learning (EARL)** to select consistent information from the simulated fire areas and multiple modalities of environmental factors to forecast/backtrack fire areas. The consistent information generally appears in reality, thus reducing the bias brought by the simulator. It allows the model training on the simulation data to leverage more realistic information. In Table 7 of the ablation study, we examine the impact of the bias of the simulated data on the model training by removing **EARL** and evaluating the S2R-FireTr model on real-world scenarios. Without **EARL**, which reduces the simulation bias, the performances of wildfire forecast and backtracking are significantly degraded.
>
> **#2. Quality of Simulation:**
>
> Here, we provide a relatively reasonable manner to approximate the quality of the simulated data. The data generated by the simulator are binary masks. In each binary mask, the pixel value (0 or 1) expresses if a coordinate on the map is located within the fire area at any given time. Based on this information, we can simulate and count the coordinates and timestamps of the fire areas within the obtained binary masks, thereby obtaining the frequency histogram of fire occurrences concerning coordinates and timestamps. Similarly, the frequency histogram can be computed based on the real-world data. We compute the Kullback-Leibler divergence (KL) and Jensen-Shannon divergence (JS) between frequency histograms of the simulated and real-world data to measure their difference concerning the spatial and temporal distribution of fire areas, for measuring the simulation quality as reported in the table below. In the last column, we also compute KL and JS between the real-world data with different sets of scenarios. Because all scenarios are taken from the real world, the differences (i.e., KL 1.93 and JS 0.15) mainly stem from the example-wise diversity between real-world scenarios. By taking the differences between the real-world scenarios as the reference, we find that the differences between the scenarios generated by each simulator and the real-world scenarios lie in KL 3.23-4.63 and JS 0.30-0.35, which are insignificant.
>
> | Simulator | FARSITE | WFDS | WRF-SFIRE | Real-world Data |
> | :---: | :---: | :---: | :---: | :---: |
> | Metrics | KL &emsp;JS | KL &emsp;JS  | KL &emsp;JS | KL &emsp;JS |
> | Real-world Data | 4.63&emsp;0.35 | 4.10&emsp;0.33 | 3.23&emsp;0.30 | **1.93&emsp;0.15** |
>
>
> **#3. Bias in Model Training:**
>
> We respectfully point out that the bias in the model training on the simulated data can be empirically evaluated by testing the trained model on real-world data. In the table below, we report the performances of our S2R-FireTr model on real-world data by training S2R-FireTr on the simulated data separately produced by the simulators FARSITE, WFDS, and WRF-SFIRE. We compare these performances to the model trained on the mixed data produced by all simulators. We use 1M simulated scenarios for model training in each case for a fair comparison. We can draw two conclusions from the following experimental data：
>
> - Compared to the training on mixed data, the model trained on data produced by a single simulator degrades performance on the forecast and backtracking tasks, with performance degradations of **16.6-24.8 AUPRC** on the forecast task and **14.3-23.4 AUPRC** on the backtracking task. The bias in the model training causes these degradations. They demonstrate the importance of using the data produced by different simulators to reduce the training bias.
> - We have also experimented with other models (e.g., ConvLSTM, UTAE, and Rainformer) in the same setting to empirically evaluate the training bias, finding the performance degradations generally more significant than **30 AUPRC** on the forecast and backtracking tasks. Our S2R-FireTr model yields slighter performance degradations, showing its strength in addressing the Sim2Real gap between model training and testing.
>
> | Task | FARSITE | WFDS | WRF-SFIRE | Mixed |
> | :---: | :---: | :---: | :---: | :---: |
> | Forecast | 48.1 | 51.7 | 56.3 | **72.9** |
> | Backtrack | 40.5 | 43.4 | 49.6 | **63.9** |

---

> ### Author Rebuttal · Authors · 2024-08-24
>
> **(1/4)**
>
> We sincerely thank you for your valuable comments.
>
> **Q1: The paper fails to provide a thorough comparison with existing similar datasets. Specific metrics and detailed evaluations are missing.**
>
> We respectfully point out that WildfireSpreadTS, Mesogeos, and other relevant datasets have been compared with the Sim2Real-Fire dataset in Table 1 of the paper. In this table, we list the number of scenarios, the coverage of the spatial range (the number of countries), tasks, the coverage of the temporal range (period), resolution, the combination of real-world and simulated data, and the number of data modalities, which are the metrics for showing the difference between datasets. Per your suggestion, we compare the Sim2Real-Fire dataset with more datasets in Table 1 of the rebuttal PDF, attached with our response to your questions. Sim2Real-Fire differs from the established datasets in the following aspects:
>
> - The Sim2Real-Fire dataset is the **first public dataset** that promotes training forest fire forecast and backtracking models on simulation data and testing them on real-world data. Our data is partially available via https://github.com/anonymous-github-Sim2Real-Fire/Sim2Real-Fire. Though some previous works share a similar idea of Sim2Real training and testing forest fire models, they do not release any data.
> - We produce the simulation data using **various simulators**, covering the empirical, numerical, and physical models commonly used in forest fire simulation. Similar works only advocate using a single simulator to generate data, leading to a significant bias in the model training.
> - The simulated fire scenarios in the Sim2Real-Fire dataset cover **wide spaces and long periods** while maintaining **more detailed** spatial and temporal resolutions than other datasets. Most of the existing datasets have coarser spatial and temporal resolutions than our dataset.
> - Every simulated scenario of the Sim2Real-Fire dataset can be sampled along the temporal dimension because it is a sequence of changing fire areas with a high temporal resolution of 1 hour (a.k.a., 1-hour time interval). This high temporal resolution allows a random temporal sampling, where a pair of adjacent fire areas can have time intervals different from others. This manner forms the sequential areas with **unequal time intervals**. It mimics the realistic situation in which the time intervals for the fire areas to be captured by satellites are unequal, allowing the models to be trained on simulated data that share similar patterns to the realistic data.

---

> ### Author Response · Authors · 2024-08-26
> **Sincerely Request Your New Comment**
>
> Dear Reviewer JYQM,
>
> We thank you again for your valuable comments, which significantly help us to polish our paper. Could we kindly know if the responses have addressed your concerns and if further explanations or clarifications are needed? Your time and efforts in evaluating our work are appreciated greatly.
>
> Best,
>
> Authors of Paper ID 141

---

> ### Comment · Reviewer_JYQM · 2024-08-28
> **Response for Rebuttal**
>
> Overall, the rebuttals for Q1 and Q2 are comprehensive and sufficiently address the points raised in the review. The additional explanations provided give confidence in the dataset's robustness and its distinct value in the context of wildfire prediction research.
>
> Also for Q3-Q5 are comprehensive and address the primary concerns raised. The additional details provided enhance clarity and justify the unique contributions of your dataset and model, demonstrating their value to the research community.
>
> the rebuttal for Q6 and Q8 is well-addressed, but Q7 little bit comment.
>
> Paper provides a solid baseline for assessing model performance. However, incorporating a comparison of your transformer-based approach with vision models could be beneficial to the machine learning community such as ViT, are widely used for handling spatial data and could serve as a relevant benchmark. By providing a comparison with these models, you could highlight the strengths and weaknesses of your approach relative to well-established methods. This would offer valuable insights and enhance the relevance of your work to a broader audience in both wildfire management and machine learning fields.
>
> I was so helpful for your rebuttal and I increased the score

---

> > ### Author Rebuttal · Authors · 2024-08-28
> >
> > Dear Reviewer JYQM,
> >
> > Again, thank you for your valuable comments, which helped us polish our paper. Below, we provide explanations for your additional questions.
> >
> > **#1. Comparison with well-established methods such as ViT**
> >
> > We respectfully clarify that the performances of ViT and S2R-FireTr have been compared in our meta-experimental data, as shown in the table below. ViT is a general transformer architecture used for single-image recognition tasks, whereas fire area forecast and backtracking require the recognition of image spatial-temporal sequences. ViT does not specifically design its model structure to cater to the spatial-temporal change of fire areas. In contrast, Rainformer, Earthformer, and Earthfarseer, which we have compared in Tables 4 and 6 of our paper, are specifically optimized transformer structures for tasks related to fire areas or other environmental factors’ spatial-temporal changes. Therefore, these models perform significantly better fire area forecast and backtracking tasks than ViT. By comparing our S2R-FireTr with Rainformer, Earthformer, and Earthfarseer, the superiority of S2R-FireTr in the crucial domain of forest fire forecast and backtracking becomes more evident. Based on your suggestion, we would be delighted to add these results to the manuscript. Minor revisions can do this to our paper.
> >
> > |Method|  |  Forecast |   |    |    Backtrack |       |
> > | :---: | :---: | :---: | :---: | :---: | :---: | :---: |
> > |    | AUPRC |  F1  |  IOU  |   AUPRC   |  F1  |  IOU
> > | ViT |61.0|69.7|51.3|56.5 | 60.4 | 38.6 |
> > | Rainformer |79.7|78.8|69.6|73.3|71.9|57.0 |
> > | Earthformer | 77.2 | 73.5 | 59.7 | 71.4 | 63.0 | 48.1|
> > | Earthfarseer | 73.4 | 70.6 | 63.5 | 69.3 | 68.5 | 50.9|
> > | S2R-FireTr | **87.3** | **83.2** |**71.2** | **78.6** | **73.5** | **58.1** |
> >
> >
> > **#2. Clarification on the final rating**
> >
> > We are pleased that your **Q1-Q6** and **Q8** have been addressed, and you have agreed with the distinct value of our work. We express our most tremendous gratitude for your increased score. We hope our above answer can solve your additional concern on **Q7**.
> >
> > We respectfully remind you that your current score is “4: Ok but not good enough - rejection”, which is still very negative. Based on this negative score, we believe you still have doubts about our work. We sincerely invite you to provide more specific feedback on our paper's shortcomings so we can better understand why you still choose to reject our article. Undoubtedly, we attach great importance to your opinion and will try to address it and improve our paper based on your suggestions.
> >
> > Best,
> >
> > Authors of Paper ID 141

---

> > > ### Comment · Reviewer_JYQM · 2024-08-29
> > > **Response by Reviewer**
> > >
> > > I think overall contribute will be effective in ml community and geo science community
> > > I changed the score.

---

> > > > ### Author Response · Authors · 2024-08-29
> > > > **Thanks for Your Review**
> > > >
> > > > Dear Reviewer JYQM,
> > > >
> > > > Thank you again for your review. We are pleased to see that the questions raised by you are solved.
> > > >
> > > > Best,
> > > >
> > > > Authors of Paper ID 141

---

### Official Review · Reviewer_T53a · 2024-07-25
**Good paper but clarifications needed**

**Rating:** 6
**Confidence:** 5
**Correctness:** Yes
**Clarity:** The paper is well-written.

**Review:**

The paper addresses an important topic as fire spread, control, and response has attracted a lot of attention from the AI/ML community in recent years. I have the following concerns:

1. It is not clear how the paper differs from prior work. For example, both WildfireDB and Mesogeos have been published at NeurIPS DB track in the last 4 years. Mesogeos is smaller than this dataset, but it incorporates several tasks, and WildfireDB specifically models fire spread and is significantly larger, i.e., it has 17 million data points.
2. Can the authors clarify what is meant by a scenario? How long does it last in time and what region does it cover in space? One potential advantage of this proposed paper over WildfireDb is that each point (of the 17m data points) is not a scenario per se, rather, each point is a daily snapshot of fire spread. If the scenarios in this paper are longer, then the authors probably have a clear advantage.
3. More importantly, in line 160, the authors simply claim that the proposed dataset is richer than others in Table 1. It is not clear why this claim stands. For example, WildfireDB and Mesogeoes incorporate vegetation, and WildfireDB incorporates topography, weather, fuel, etc.
4. The dataset and the proposed algorithmic approach are orthogonal. The proposed approach could be tried on the other datasets as well; however, without clarifying how the dataset is novel and different from prior work, the contribution of the benchmarking approach is rather limited.

**Strengths:**

Please see above.

**Additional Feedback:**

NA

**Documentation:**

Sufficient.

**Limitations:**

Please see above.

**Opportunities For Improvement:**

Please see above. I am open to changing my score based on the rebuttal.

I have read through the rebuttal which answers my questions. I have updated my score.

**Relation To Prior Work:**

Not clear at all. Please address the comments above.

**Summary And Contributions:**

The paper presents 1 million simulation scenarios for forest fires and a model for forecasting and backtracking the spread of wildfire.

---

> ### Author Rebuttal · Authors · 2024-08-12
>
> We sincerely thank you for your valuable comments.
>
> **Q1: How does the paper differ from prior works?**
>
> This paper differs from the prior works in the following aspects:
>
> - The Sim2Real-Fire dataset is the **first public dataset** that promotes training forest fire forecast and backtracking models on simulation data and testing them on real-world data. Our data is partially available via https://github.com/anonymous-github-Sim2Real-Fire/Sim2Real-Fire. Though some previous works share a similar idea of Sim2Real training and testing forest fire models, they do not release any data.
> - We produce the simulation data using **various simulators**, covering the empirical, numerical, and physical models commonly used in forest fire simulation. Similar works only advocate using a single simulator to generate data, leading to a significant bias in the model training.
> - The simulated fire scenarios in the Sim2Real-Fire dataset cover **wide spaces and long periods** while maintaining **more detailed** spatial and temporal resolutions than other datasets. Please see **Q2** for comparison with other datasets.
> - Our models can predict and backtrack fire areas with **unequal time intervals**. In contrast, previous methods only assume the interval is equal (e.g., 1 day). They fall short in predicting and backtracking fire areas that are not observed by satellites in reality. Please see **Q4** for the special design of our models.
>
>
> **Q2: What is meant by a scenario? How long does it last in time? what region does it cover in space?**
>
> Figure 1 of the paper shows that each scenario contains sequential satellite images of fire areas and weather information. The sequential data lasts about 10 days. The time interval between adjacent frames is 1 hour. It contains the spatial data of topography, vegetation, and fuel within 20km$^2$. The spatial resolution is 30m$\times$30m=900m$^2$. In comparison, the scenario in WildfireDB is only captured at a particular moment rather than a sequence. The spatial and temporal resolution of Mesogeos is 1km$^2$ $\times$1 day, which is coarser than 900m$^2$ $\times$1 hour of our Sim2Real-Fire.
>
> WildfireDB and Mesogeos cover the USA and the Mediterranean regions, respectively, spanning about 9,000,000km$^2$. Sim2Real-Fire covers the USA, Canada, and Mexico, spanning about 20,000,000km$^2$. It allows us to consider wider regions' vegetation, topography, and weather, producing forest fire scenarios with more diverse patterns. We will add the above details to the paper.
>
> **Q3: Why is the proposed dataset richer than others in Table 1?**
>
> In response to your **Q2**, we have pointed out that the Sim2Real-Fire dataset provides a long time and broad space of forest fire scenarios while maintaining more detailed temporal and spatial resolutions than other datasets like WildfireDB and Mesogeoes. Though WildfireDB and Mesogeoes also provide vegetation, topography, and fuel maps, the long time, broad space, and high resolution mainly contribute to the richness of the spatial and temporal data of Sim2Real-Fire.
>
>
> **Q4: The dataset and the proposed algorithmic approach are orthogonal.**
>
> We contribute the Sim2Real-Fire dataset to the research community to promote training forest fire forecast and backtracking models on simulation data and to test them on real-world data. This research setting faces the challenge of the Sim2Real gap (as described in lines 55-64) and temporally incomplete satellite data (lines 42-46). Thus, we propose the deep transformer network, S2R-FireTr, to forecast and backtrack the forest fire. Tables 2, 4, and 6 of the paper and Table 1 of the supplementary file compare S2R-FireTr with the general deep networks like CNN, LSTM, and Transformer, where S2R-FireTr achieves better results in this setting.
>
> First, S2R-FireTr has the transformer-based component of **Environment-guided Area Representation Learning (EARL)**. During the model training on the simulation data, this component uses cross attention to take the simulated fire areas as the query, while the environmental information (i.e., topography, vegetation, fuel, and weather) plays the key and value. **EARL** selects consistent information from the simulated fire areas and multiple modalities of environmental factors to forecast/backtrack fire areas. The consistent information generally appears in reality, thus allowing the model training on the simulation data to leverage more realistic information. Typically, the general deep networks mix the simulated fire areas and the environmental information as input without considering the information selection like S2R-FireTr.
>
> S2R-FireTr has another transformer-based component of **Time-wise Fire Area Regression (TFAR)**, which can forecast and backtrack fire areas at arbitrary moments. This is done by adding a special design of timestamps embedding into the cross attention. These timestamps have unequal temporal intervals. TFAR better considers the temporal incompletion of fire areas, which may be captured by different satellites flying over at unequal time intervals. For more general purposes, the deep networks assume the time intervals between sequential frames are equal, thus lagging behind S2R-FireTr.
>
>
> **Q5: The proposed approach could be tried on the other datasets.**
>
> The tables below compare S2R-FireTr with LSTM and transformer on the WildfireSpreadTS dataset. Here, the top and bottom tables represent the tasks of forecasting a single and a sequence of fire areas, where S2R-FireTr outperforms other methods.
>
>
> | Method | AUPRC | F1 |  IOU |
> | :---: | :---: | :---: | :---: |
> | ConvLSTM | 31.9 | 55.4 | 25.3 |
> | UTAE | 38.8 | 59.6 | 28.4 |
> | Rainformer  |  40.7  | 61.2 |  31.0 |
> | S2R-FireTr  |  **44.1**  | **66.9**  |  **35.7** |
>
> | Method | AUPRC | F1 |  IOU |
> | :---: | :---: | :---: | :---: |
> | ConvLSTM | 15.2 | 23.4 | 11.7 |
> | UTAE | 16.5 | 28.7 | 13.0 |
> | Rainformer  |  18.1  | 34.0 |  14.8 |
> | S2R-FireTr  |  **20.3**  | **35.2**  |  **16.1** |

---

> > ### Comment · Reviewer_T53a · 2024-08-19
> > **Reviewer Response**
> >
> > I appreciate the detailed response. I have a few additional questions:
> >
> > 1. "Though some previous works share a similar idea of Sim2Real training and testing forest fire models, they do not release any data.": What works are these? Several sources (e.g., WildfireDB and others highlighted by reviewer JYQM22 have all of their data openly released). There is some confusion here. Can you point out what prior work you are referring to?
> >
> > 2. "leading to a significant bias in the model training." I understand this claim. How was this empirically evaluated?
> >
> > 3. In line with what JYQM22 has raised, how were the quality of simulations gauged?

---

> > > ### Author Rebuttal · Authors · 2024-08-24
> > >
> > > Dear Reviewer T53a,
> > >
> > > We thank you again for your valuable comments.
> > >
> > > **Q1: Several sources (e.g., WildfireDB and others have all of their data openly released). There is some confusion here. Can you point out what prior work you are referring to?**
> > >
> > > We are sorry for missing the references, which have been added below. These works only share similar ideas of Sim2Real training and testing forest fire models without releasing the combination of simulated and real-world data like our Sim2Real-Fire dataset for the research of Sim2Real model training and testing. Note that WildfireDB and relevant works highlighted by reviewer JYQM **DO NOT** share any simulated data for facilitating the Sim2Real model training and testing for forest fire, which our paper advocates. Based on this, WildfireDB and relevant works propose the forest fire analysis models without considering the impact of the simulated data. We provide an extra PDF with this response to compare to the existing datasets.
> > >
> > > - Wildland fire spread modeling using convolutional neural networks, Fire technology, 2019.
> > > - A spatio-temporal neural network forecasting approach for emulation of firefront models, Signal Processing, 2022.
> > > - Recurrent convolutional deep neural networks for modeling time-resolved wildfire spread behavior, Fire technology, 2023.
> > >
> > >
> > > **Q2: How was bias empirically evaluated?**
> > >
> > > We respectfully point out that the bias in the model training on the simulated data can be empirically evaluated by testing the trained model on real-world data. In the table below, we report the performances of our S2R-FireTr model on real-world data by training S2R-FireTr on the simulated data separately produced by the simulators FARSITE, WFDS, and WRF-SFIRE. We compare these performances to the model trained on the mixed data produced by all simulators. We use 1M simulated scenarios for model training in each case for a fair comparison. We can draw two conclusions from the following experimental data：
> > >
> > > - Compared to the training on mixed data, the model trained on data produced by a single simulator degrades performance on the forecast and backtracking tasks, with performance degradations of **16.6-24.8 AUPRC** on the forecast task and **14.3-23.4 AUPRC** on the backtracking task. The bias in the model training causes these degradations. They demonstrate the importance of using the data produced by different simulators to reduce the training bias.
> > > - We have also experimented with other models (e.g., ConvLSTM, UTAE, and Rainformer) in the same setting to empirically evaluate the training bias, finding the performance degradations generally more significant than **30 AUPRC** on the forecast and backtracking tasks. Our S2R-FireTr model yields slighter performance degradations, showing its strength in addressing the Sim2Real gap between model training and testing.
> > >
> > > | Task | FARSITE | WFDS | WRF-SFIRE | Mixed |
> > > | :---: | :---: | :---: | :---: | :---: |
> > > | Forecast | 48.1 | 51.7 | 56.3 | **72.9** |
> > > | Backtrack | 40.5 | 43.4 | 49.6 | **63.9** |
> > >
> > > **Q3: How were the quality of simulations gauged?**
> > >
> > > In response to **Q2** of Reviewer JYQM, we have pointed out that building a simulated dataset with high quality, variability, and fidelity levels but without bias is critical for training wildfire recognition models. As we have discussed in lines 55-64 of the paper, the quality and bias of the simulated data are challenging to measure in reality, and they should be carefully studied in more focused research. Below, we provide a relatively reasonable manner to approximate the quality of the simulated data.
> > >
> > > Please note that the data generated by the simulator are binary masks. In each binary mask, the pixel value (0 or 1) expresses if a coordinate on the map is located within the fire area at any given time. Based on this information, we can simulate and count the coordinates and timestamps of the fire areas within the obtained binary masks, thereby obtaining the frequency histogram of fire occurrences concerning coordinates and timestamps. Similarly, the frequency histogram can be computed based on the real-world data. We compute the Kullback-Leibler divergence (KL) and Jensen-Shannon divergence (JS) between frequency histograms of the simulated and real-world data to measure their difference concerning the spatial and temporal distribution of fire areas, for measuring the simulation quality as reported in the table below. In the last column, we also compute KL and JS between the real-world data with different sets of scenarios. Because all scenarios are taken from the real world, the differences (i.e., KL 1.93 and JS 0.15) mainly stem from the example-wise diversity between real-world scenarios. By taking the differences between the real-world scenarios as the reference, we find that the differences between the scenarios generated by each simulator and the real-world scenarios lie in KL 3.23-4.63 and JS 0.30-0.35, which are insignificant.
> > >
> > > | Simulator | FARSITE | WFDS | WRF-SFIRE | Real-world Data |
> > > | :---: | :---: | :---: | :---: | :---: |
> > > | Metrics | KL &emsp;JS | KL &emsp;JS  | KL &emsp;JS | KL &emsp;JS |
> > > | Real-world Data | 4.63&emsp;0.35 | 4.10&emsp;0.33 | 3.23&emsp;0.30 | **1.93&emsp;0.15** |
> > >
> > > Rather than using tricky methods to work around the issues of low quality and bias in the dataset, this paper focuses on pushing the limit of using the simulated data with potentially low-quality and biased information for training. To reduce the impact of low-quality and biased information for model training, our Sim2Real dataset contains variant data produced by various simulators. Furthermore, our S2R-FireTr can select consistent information from the simulated fire areas and multiple modalities of environmental factors to forecast/backtrack fire areas. Many thanks if the above point can be considered to justify the value of our paper.

---

> ### Author Response · Authors · 2024-08-26
> **Sincerely Request Your New Comment**
>
> Dear Reviewer T53a,
>
> We thank you again for your valuable comments, which significantly help us to polish our paper. Could we kindly know if the responses have addressed your concerns and if further explanations or clarifications are needed? Your time and efforts in evaluating our work are appreciated greatly.
>
> Best,
>
> Authors of Paper ID 141

---

> > ### Comment · Reviewer_T53a · 2024-08-28
> > **Update**
> >
> > I have gone through the answers and updated my score.

---

> > > ### Author Response · Authors · 2024-08-29
> > > **Thanks for Your Review**
> > >
> > > Dear Reviewer T53a,
> > >
> > > Thank you again for your review. We are pleased to see that the questions raised by you are solved.
> > >
> > > Best,
> > >
> > > Authors of Paper ID 141

---

### Official Review · Reviewer_Zjdq · 2024-07-28
**Review of Sim2Real-Fire**

**Rating:** 8
**Confidence:** 4
**Correctness:** Yes
**Clarity:** Yes

**Review:**

The paper presents a high-quality, original, and significant contribution to the field of wildfire prediction, with some areas for improvement in clarity and detail.

**Strengths:**

The paper is well-written and well-organized. The figures and tables are very nice.

The paper presents a comprehensive dataset Sim2Real-Fire, which is substantial in size and includes multi-modal data.

The methodology for creating and validating the dataset is robust, with both simulated and real-world data.

The proposed model, S2R-FireTr, significantly improves over state-of-the-art methods in wildfire prediction tasks.

Detailed experimental results and comparisons are provided, demonstrating the effectiveness of the proposed approach.

**Additional Feedback:**

-

**Documentation:**

Based on the authors' GitHub repository, the documentation can be much improved in the following aspects:

Detailed Dataset Description is missing,
Metadata: Detailed metadata for each dataset file, including format specifications and data types.
Annotations: Information about how the annotations were performed, including the criteria and tools used
Data Preprocessing Steps:
Instructions: Clear instructions on how to preprocess the data before training the model.

**Usage Examples**
Tutorials: Step-by-step tutorials or Jupyter notebooks demonstrating how to use the dataset for training and testing models.
Code Snippets: Examples of code snippets for loading, manipulating, and visualizing the dataset.

Evaluation Metrics:
Definitions: Definitions of the evaluation metrics used (e.g., AUPRC, F1, IOU) and how they are calculated.

**Limitations:**

Yes in the supplement, but not in the main paper.

**Opportunities For Improvement:**

The clarity of some technical descriptions, particularly around the architecture and specific methods used, the cross-attention mechanism could be improved.

The dataset creation process might benefit from additional details on data preprocessing and integration steps.

The paper could discuss more about potential limitations or biases in the dataset and how they might affect the model's performance.

**Relation To Prior Work:**

The analysis with previous efforts could be better. In particular, though the integration of simulated and real-world data is innovative, the paper could better explain how this approach compares to previous efforts.

**Summary And Contributions:**

The paper introduces the Sim2Real-Fire dataset, comprising 1 million simulated wildfire scenarios and 1,000 real-world wildfire scenarios, designed to train AI models for wildfire forecasting and backtracking. The dataset includes multi-modal environmental information such as topography, vegetation, fuel maps, weather data, and satellite images. The authors also propose a deep transformer network, S2R-FireTr, which outperforms state-of-the-art methods in real-world wildfire scenarios.

---

> ### Author Rebuttal · Authors · 2024-08-17
>
> **(2/2)**
>
> **Q3: The paper could discuss more about potential limitations or biases in the dataset and how they might affect the model's performance.**
>
> Thanks for your advice. Suppose a data set can cover as much information as possible about the worldwide topography, vegetation, fuel, weather conditions, etc.. In that case, its training data can effectively improve the generalization of fire area forecast and backtracking models. Based on this ideal intuition, we discuss the limitation and potential opportunity brought by our Sim2Real-Fire dataset:
>
> - **Limitation**. Although the Sim2Real-Fire dataset has unique attributes (see our response to **Q1**), it is a closed set like most existing datasets. Sim2Real-Fire only includes wildfire scenarios from certain countries and periods due to the limited budget for data acquisition in reality. It significantly reduces the richness of the training data. The model, trained on the data with a few known categories of topography, vegetation, fuel, and weather conditions, eventually lacks the generalization power to forecast the fire areas under unknown environmental conditions.
> - **Opportunity**. As we have stated in **Limitation**, it is inevitable that closed datasets cannot provide comprehensive environmental information. Therefore, a cost-effective and fast data acquisition method must be considered to dynamically add more data and change the closed set to the open dataset. Some existing datasets, like MODIS Thermal Anomaly, VIIRS Thermal Anomaly, GOES Wildfire, NOAA HMS, and NIFC Wildfire Perimeters, are open to refresh the environmental data. However, these datasets rely on continuous monitoring by satellites, leading to high data acquisition costs. In addition, there is a large amount of redundant data without fire areas, thus posing difficulties for data cleaning and model training. This paper advocates minimizing the Sim2Real gap between the simulated and real-world scenarios, which is the core of our paper apart from the dataset and methodology contributions. **Minimizing the Sim2Real gap is challenging, but once we can address it, we can use cost-effective simulators to produce a large amount of data rapidly**. By combining different environmental conditions, we can produce large-scale data to add to the Sim2Real-Fire dataset, which becomes an open set for model training. This fashion lets the model witness simulated fire areas under different environmental conditions that probably appear in real-world scenarios, where the generalization power of the model is enhanced.
>
> Actually, we propose the transformer-based component **EARL** of S2R-FireTr to reduce the simulation bias. **EARL** selects consistent information from the simulated fire areas and multiple modalities of environmental factors to forecast/backtrack fire areas. The consistent information generally appears in reality, thus reducing the bias brought by the simulator. It allows the model training on the simulation data to leverage more realistic information. In Table 7 of the ablation study, we examine the impact of the bias of the simulated data on the model training by removing **EARL** and evaluating the S2R-FireTr model on real-world scenarios. Without **EARL**, which reduces the simulation bias, the performances of wildfire forecast and backtracking are significantly degraded.
>
>
> **Q4: Detailed Dataset Description is missing, Metadata: Detailed metadata for each dataset file, including format specifications and data types. Annotations: Information about how the annotations were performed, including the criteria and tools used Data Preprocessing Steps: Instructions: Clear instructions on how to preprocess the data before training the model.**
>
> Thanks for your advice. In Table 1  of the rebuttal PDF, we describe the metadata that will be used to reorganize the GitHub repository. The annotations and data preprocessing have been described in our response to **Q2**.
>
> **Q5: Definitions of the evaluation metrics used (e.g., AUPRC, F1, IOU) and how they are calculated.**
>
> We respectfully point out that AUPRC, F1, and IOU are the most commonly used metrics to evaluate the accuracy of pixel-level image area recognition.
>
> - **AUPRC** is a metric used to measure the trade-off between Precision and Recall for different threshold values. It provides a single scalar value that represents the area under the Precision-Recall curve. AUPRC is formulated as:
>
> $\text{AUPRC} = \int_{0}^{1} \text{Precision}(\text{Recall}) \~d(\text{Recall}),$
> where
> $\text{Precision} = \frac{\text{True Positives}}{\text{True Positives} + \text{False Positives}}$,
> $\text{Recall} = \frac{\text{True Positives}}{\text{True Positives} + \text{False Negatives}}$.
>
> - **F1 score** is the harmonic mean of Precision and Recall, providing a balance between the two metrics. Its formulation is:
>
> $\text{F1 Score} = 2 \times \frac{\text{Precision} \times \text{Recall}}{\text{Precision} + \text{Recall}}$.
>
> - **IoU** measures the overlap between the predicted region and the ground truth region. It is defined as the ratio of the intersection area over the union area of the predicted and actual regions. Its formulation is:
>
> $\text{IoU} = \frac{\text{Intersection}}{\text{Union}} = \frac{\text{True Positives}}{\text{True Positives} + \text{False Positives} + \text{False Negatives}}$.

---

> ### Author Rebuttal · Authors · 2024-08-17
>
> **(1/2)**
>
> We sincerely thank you for your valuable comments.
>
> **Q1: The clarity of some technical descriptions, particularly around the architecture and specific methods used, the cross-attention mechanism could be improved. Though the integration of simulated and real-world data is innovative, the paper could better explain how this approach compares to previous efforts.**
>
> Thanks for your suggestion, which helps us polish our technical details presentation. Here, we are glad to highlight the architecture and specific designs of our S2R-FireTr model by comparing it to previous efforts. S2R-FireTr is a deep transformer network for forecasting and backtracking forest fires. The model is trained on the simulated fire scenarios and tested on the real-world scenarios. S2R-FireTr has two specific components to reduce the Sim2Real gap between the simulated and real-world scenarios (the definition of the Sim2Real gap can be found in lines 55-64 of the paper) and the temporally incomplete data (see lines 42-46):
>
> - S2R-FireTr has the transformer-based component of **Environment-guided Area Representation Learning (EARL)**. During the model training on the simulation data, this component uses cross attention to take the simulated fire areas as the query. Environmental information (i.e., topography, vegetation, fuel, and weather) plays the key and value. **EARL** selects consistent information from the simulated fire areas and multiple modalities of environmental factors to forecast/backtrack fire areas. The consistent information generally appears in reality, thus allowing the model training on the simulation data to leverage more realistic information. **Typically, the general deep networks mix the simulated fire areas and the environmental information as input without considering the information selection like S2R-FireTr**.
> - S2R-FireTr has another transformer-based component of **Time-wise Fire Area Regression (TFAR)**, which can forecast and backtrack fire areas at arbitrary moments. This is done by adding a special design of timestamps embedding into the cross attention. These timestamps have unequal temporal intervals. Specifically, we embed these timestamps (see Section A of the supplementary file) into the latent representations input into the cross-attention with the temporal and spatial representations from **EARL**. This procedure is illustrated in Figure 3. This is how the model handles the temporal incomplete data. It allows TFAR to better consider the temporal incompletion of fire areas, which may be captured by different satellites flying over at unequal time intervals. **For more general purposes, the deep networks assume the time intervals between sequential frames are equal, thus lagging behind S2R-FireTr.**
>
> Tables 2, 4, and 6 of the paper and Table 1 of the supplementary file compare S2R-FireTr with the general deep networks like CNN, LSTM, and Transformer, where S2R-FireTr achieves better results.
>
>
> **Q2: The dataset creation process might benefit from additional details on data preprocessing and integration steps.**
>
> We plan to add details about data selection and integration from the following aspects:
>
> - **Wildfire localization**. We selected the fire data from 2013 to 2023 in the United States, Canada, and Mexico from the Global Wildfire Database and recorded the longitude and latitude of fire occurrences.
> - **Environmental information association**. Based on the latitudes and longitudes obtained from the Global Wildfire Database, we retrieved the corresponding topography, vegetation, and fuel information for each fire scenario from the Landfire website, as well as the corresponding weather information from the Remote Automatic Weather Station website and meteomanz.com.
> - **Production of simulated fire**. We input the environmental information into different simulators (i.e., FARSITE, WFDS, and WRF-SFIRE), producing the masks of the changing fire areas.
> - **Pre-processing and annotation of real-world fire**. We exclude satellite images that fail to clearly depict fire regions owing to the obstruction caused by dense clouds or smoke. Next, we upload the satellite data into the NV5 GEOSPATIAL platform, where we select the Radiometric Calibration tool from the comprehensive toolbox to procure a precisely calibrated radiance image. Following this, we employ the Quick Atmospheric Correction feature to acquire a FLAASH atmospheric correction image, ensuring utmost accuracy. To maintain consistency and capture the intricate landscapes of Earth at a resolute 30-meter spatial resolution, we utilize the Resample tool within the ArcGIS toolbox. The preprocessed satellite images are imported into ArcGIS for manual annotation. After three individuals simultaneously annotate a single satellite image following the aforementioned method, their output results are compared, and discrepancies in annotations are further verified. The final annotated results are then outputted.
>
> We have provided a video on the homepage of our GitHub repository to illustrate the above steps more directly.

---

> ### Author Response · Authors · 2024-08-26
> **Sincerely Request Your New Comment**
>
> Dear Reviewer Zjdq,
>
> We thank you again for your valuable comments, which significantly help us to polish our paper. Could we kindly know if the responses have addressed your concerns and if further explanations or clarifications are needed? Your time and efforts in evaluating our work are appreciated greatly.
>
> Best,
>
> Authors of Paper ID 141

---

> > ### Author Response · Authors · 2024-08-29
> > **Sincerely Request Your New Comment Again**
> >
> > Dear Reviewer Zjdq,
> >
> > Again, please allow us to extend our sincere thanks to you, for your time and efforts of reviewing our paper. As the deadline for the authors' response is approaching, we sincerely request your comment on our primary response. This will definitely give us a valuable chance to address the questions unsolved.
> >
> > Best,
> >
> > Authors of Paper ID 141

---

### Author Rebuttal · Authors · 2024-08-28

Dear Reviewers,

Again, please allow us to extend our sincere thanks for your time and efforts in reviewing our paper. As the authors' response deadline is approaching, we sincerely request your comments on our responses. This will definitely give us a valuable chance to address the unresolved questions.

Best,

Authors of Paper ID 141

---

### Decision · Program_Chairs · 2024-09-26

**Decision:**

Accept (Poster)

**Comment:**

This paper has three positive scores finally (8, 6, 6). It provides large-scale, multi-modal simulation data in the Forest Fire Forecast and Backtracking field with various scenarios. In the rebuttal period, the authors respond well to the reviewers' concerns on the differences and advantages from previous work, the characteristics and quality of their benchmark, potential dataset biases, evaluation metrics, and model details. Based on the authors' detailed response, two reviewers (Reviewer JYQM and T53a) raised their scores.
AC considers this benchmark would contribute to developing and evaluating the Forest Fire Forecast and Backtracking field. Therefore,  accept  is recommended to this paper. The authors are recommended revise the paper based on the concerns raised by the reviewers in the rebuttal period.